# Organic and inorganic sublattice coupling in two-dimensional lead halide perovskites

Jianhui Fu ®[1], Tieyuan Bian[2], Jun Yin ®[2] ✉, Minjun Feng ®[1], Qiang Xu[1], Yue Wang ®[1] & Tze Chien Sum ®[1] ✉

Two-dimensional layered organic-inorganic halide perovskites have successfully spread to diverse optoelectronic applications. Nevertheless, there remain gaps in our understanding of the interactions between organic and inorganic sublattices that form the foundation of their remarkable properties. Here, we examine these interactions using pump-probe spectroscopy and ab initio molecular dynamics simulations. Unlike off-resonant pumping, resonant excitation of the organic sublattice alters both the electronic and lattice degrees of freedom within the inorganic sublattice, indicating the existence of electronic coupling. Theoretical simulations verify that the reduced bandgap is likely due to the enhanced distortion index of the inorganic octahedra. Further evidence of the mechanical coupling between these two sublattices is revealed through the slow heat transfer process, where the resultant lattice tensile strain launches coherent longitudinal acoustic phonons. Our findings explicate the intimate electronic and mechanical couplings between the organic and inorganic sublattices, crucial for tailoring the optoelectronic properties of two-dimensional halide perovskites.

Over the past few years, there has been a growing fascination with layered two-dimensional (2D) organic-inorganic lead halide perovskites due to their remarkable optoelectronic properties, facile fabrication process and robust chemical stability. They hold great promise for a plethora of optoelectronic device applications, particularly for photodetectors[1], lasers[2], field effect transistors[3], and light emitting diodes[4]. Unlike their three-dimensional (3D) counterparts, layered 2D hybrid lead halide perovskites, with a chemical formula of $A_2PbX_4$ (where A is a monovalent organic cation, X is a halide ion), comprises of one sheet of corner-sharing $PbX_6$ octahedra. These octahedra are sandwiched by bilayers of bulkier organic spacers (Fig. 1a) that are in turn held together by van der Waals forces[5,6] similar to 2D van der Waals heterostructures[7] and MXenes[8]. The presence of the large organic cation plays a significant role in the structural, optical, mechanical, thermal, and electronic properties of 2D halide perovskites[9–14]. The low dielectric constant of the organic moiety interlayer, as compared to the inorganic layer, contributes significantly

to the dielectric confinement, which together with the quantum confinement, enables a sizable oscillator strength and a large exciton binding energy (~200–300 meV) for 2D halide perovskites[15,16]. Meanwhile, organic cations with varying lengths can alter the interlayer distance between the inorganic layer, thereby modifying both the quantum confinement and the oscillator strength[17]. While intramolecular interactions within the organic layers are found to contribute to the structural phase sequence of the crystals[18,19] and low carrier mobility[20], the gauche defect and local chain distortion in organic cations have been proposed as the cause of the trap-induced broad band-tail emission[20] in 2D halide perovskites.

Despite significant progress in understanding the role of the organic cation, much less is known about the interactions between the organic cation and the inorganic octahedra sublattice. While it is well-established that these two sublattices are connected to one another via hydrogen bonding and electrostatic interactions, a study conducted by Guo et al. suggests that these two sublattices in 3D halide

[1]Division of Physics and Applied Physics, School of Physical and Mathematical Sciences, Nanyang Technological University, 21 Nanyang Link, Singapore 637371, Singapore. [2]Department of Applied Physics, The Hong Kong Polytechnic University, Kowloon, Hong Kong 999077, PR China. ✉e-mail: jun.yin@polyu.edu.hk; Tzechien@ntu.edu.sg

perovskites are electronically decoupled except for their thermal properties. Nevertheless, another study paints a different picture that the biexciton binding energy is reduced by the dynamic disorder due to the vibrational motion of the organic moiety[21]. The coupling between the different nuclear degrees of freedom influences the lattice anharmonicity and indirectly affects the electron–phonon interactions, where the latter plays a crucial role in the properties of 2D halide perovskites, such as hot carrier cooling[22], PL linewidth[23], and carrier mobility[24], etc. Hence, elucidating the interplay between these two sublattices is not only fundamentally important for a clear understanding but also essential for optimizing the performance of 2D halide perovskites-based optoelectronic devices.

Herein, we scrutinize the organic and inorganic sublattice coupling by directly exciting the organic sublattice with femtosecond mid-infrared (MIR) laser pulses, followed by probing the electronic transition of the inorganic sublattice with a white light continuum probe pulse. The electronic coupling between these two sublattices contributes to the modified lattice degree of freedom of the inorganic sublattice which is associated with a decrease of the optical bandgap arising most likely from an enhanced distortion index of the inorganic sublattice. Meanwhile, the mechanical coupling between these two sublattices contributes to a slow heat transfer process, resulting in the generation of thermally driven-lattice strain that launches the soft coherent longitudinal acoustic phonons (CLAPs). Our study clarifies the interplay between the organic and inorganic sublattice couplings crucial for engineering the optoelectronic properties of 2D hybrid halide perovskites. This understanding may have significant implications in the field of nonlinear phononics[25].

## Results

### Mid-infrared pump−visible probe measurements

Archetypical 2D perovskite phenylethyl ammonium lead iodide ($(PEA)_2PbI_4$) films are prepared using spin-coating. Details of sample preparation and characterizations are provided in the Methods section and Supplementary Figs. 1, 2. We disentangle the inorganic-organic sublattice coupling in $(PEA)_2PbI_4$ by selectively exciting the organic PEA sublattice with a MIR pulse and subsequently detecting the electronic transitions in the inorganic lattice using a white-light continuum probe pulse (Fig. 1a). This is possible given that: (1) the band-edge density of states of $(PEA)_2PbI_4$ mainly consists of valence states from the inorganic $PbI_6$ octahedra; and (2) the organic PEA cation only plays an indirect role on the density of states through its cation size and the hydrogen bonding which connects the organic and inorganic sub-lattices that in turn affects the octahedral distortion and spin-orbit coupling[26–28]. Therefore, the MIR laser pulse only excites the organic PEA cation, while electronic transitions from the inorganic $PbI_6$ sublattice remain largely unaffected.

Figure 1b shows the good overlap between the MIR laser pulse profile and the room temperature (RT) infrared (IR) absorption spectrum of $(PEA)_2PbI_4$ around $3000\,cm^{-1}$ (dominated by the N−H stretching vibrational motion)[29], thus permitting resonant excitation of the PEA cation. Figure 1c shows the transient absorption (TA) spectrum ($\Delta A = A - A_0$, where $A$ and $A_0$ are the absorbance of the sample with and without MIR pump), and linear absorption spectrum of $(PEA)_2PbI_4$ films at 77 K. As demonstrated in our previous work[11], the linear absorption spectrum comprises of two peaks due to transitions from a bright free exciton state (EX) at around 524 nm and its vibronic coupling state (EX + Δ, where Δ is the phonon energy) at around 519 nm arising from strong exciton coupling to the vibrational motion of PEA cation[30,31]. Therefore, the TA spectrum exhibits responses from these two states. However, different from above-bandgap photoexcitation[11] (i.e., visible pump pulse), the TA spectrum for resonantly excited PEA cations (i.e., MIR pump pulse) shows a complex evolution of positive and negative ΔA bands with time delay. Figure 1d shows the TA kinetics of P1, which exhibits four distinct regimes: regime ① with a sharp negative P1 that occurs within the pump-probe overlap; regime ② in the first ~10 ps with

a positive P1 which increases with delay; regime ③ where the amplitude changes from positive to negative on the order of hundreds of ps; and regime ④ with a negative P1 on a nanosecond timescale whose amplitude decreases with delay. Representative TA spectra at different time delays of these regimes can be seen in Fig. 1e. The TA spectral evolution following resonant excitation of PEA cations can be understood based on the change in electronic transition energy. Figure 1f shows that a blue-shift of the electronic transition results in a negative P1 whereas a red-shift leads to a positive P1. The change in electronic transition energy can thus be evaluated by monitoring ΔA of P1.

We assign regime ① to the optical stark effect-induced increase of exciton energy because the TA spectrum consisting of two photobleaching (PB, $\Delta A < 0$) bands and two photoinduced absorption (PIA, $\Delta A > 0$) bands at the red and blue sides of the PB bands (Supplementary Fig. 3) is similar to that caused by the band-filling effect. We attribute regime ② to the intramolecular vibrational energy redistribution (IVER) of the PEA cation in which sequential energy downconversion takes place through all the intermediate phonon modes starting from the excited N−H stretching mode to the lowest-lying phonon modes. The timescale of this process normally involves all the intermediate modes and well matches previous MIR pump-visible probe measurements in 2D halide perovskites[32] as well as the $CH_2I_2$ and $CH_3I$ molecules[33,34]. Concomitant with this IVER process is the generation of transient micro-strain which will distort the $PbI_6$ octahedra, resulting in a decrease of exciton energy due to a positive bandgap deformation potential[9,35]. We note that a similar phenomenon was recently reported for the transient reflection of butylammonium lead bromide ($(BA)_2PbBr_4$) perovskite with a MIR pump, which is attributed to the reduction of exciton oscillator strength arising from expansion of photoexcited BA cation[32]. Nevertheless, such assignment is based on the assumption that the interaction between organic cation and Pb−Br network in this regime is negligible and therefore the inorganic $PbBr_6$ octahedra did not undergo any distortion. This in fact may not be the case, since the stretching motion of the N−H bond will affect the hydrogen bonding between H and Br atoms, and hence the distortion of $PbBr_6$ octahedra and the exciton energy. On the other hand, from the physical point of view, the reduction in oscillator strength corresponds to the amplitude modulation of the absorption spectrum, the resultant ΔA (Supplementary Fig. 4) is however, inconsistent with our experimental results (Fig. 1e). Therefore, we attribute regime ② to the IVER of the PEA cation, which is associated with the distortion of $PbI_6$ octahedra and the reduction of optical bandgap. Similar change of optical bandgap by direct excitation of $PbI_6$ octahedra vibrational motion using THz pump was observed in $MAPbI_3$ films[36].

Regime ③ can be assigned to the heat transfer process where the vibrational energy of the thermalized PEA cation is transferred to the phonon modes of $PbI_6$ octahedra that will eventually reach a quasi-equilibrium state amongst themselves. In this process, the population of high-energy phonon modes is reduced which is associated with the population increase of low-energy phonon modes. Consequently, the whole lattice will be heated up, resulting in a lattice expansion which is associated with the increase of exciton transition energy, thereby generating a band-filling-like TA spectrum[37]. The thermal lattice expansion-induced exciton energy increase is consistent with the temperature dependence of the optical bandgap of $(PEA)_2PbI_4$, which possesses a positive bandgap deformation potential[9,38]. Similar slow heat transfer process from the relaxation of vibrational motion of the organic lattice to that of the inorganic lattice has been reported in $MAPbI_3$ films[37] and colloidal CdSe nanocrystals[39]. In contrast with conventional phonon relaxation process (such as intramolecular vibrational energy relaxation[40] and phonon-phonon equilibration in most semiconductors[22,41]) due to anharmonic phonon-phonon interactions which in general takes place within several picoseconds[22], this slow heat transfer process arising from interactions between different sublattices is two orders of magnitude slower. This may be because (1)

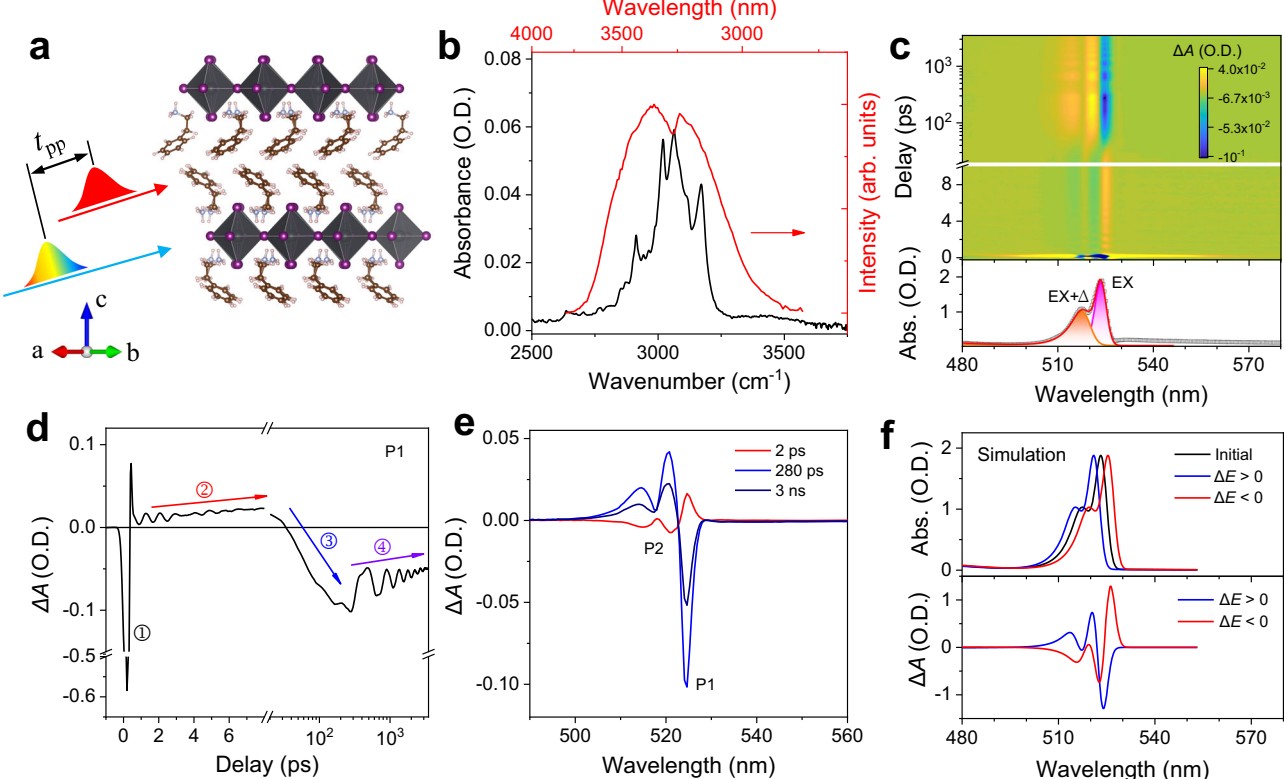

**Fig. 1 | Observation of optical bandgap shift in (PEA)₂PbI₄ films by resonant excitation of PEA cation using a MIR laser pulse centered at 3.3 μm. a** Schematic of resonant excitation of PEA cations using a MIR laser pump pulse and subsequent visible probe of inorganic lattice. $t_{pp}$ refers to the pump-probe delay time. **b** Spectral profile of the 3.3 μm MIR laser pulse (red) and FTIR absorption spectrum (black) of the film at RT. **c** TA spectrum as compared to linear absorption spectrum and the curve fits of the films at 77 K. In the TA measurement, the films were pumped at 3.3 μm with a fluence of 2 mJ cm⁻². **d** TA kinetics probed at P1 on the long timescale showing four regimes of different processes. **e** Representative TA spectrum of the films at delays of 2280, and 3000 ps. **f** Simulated absorption spectra when the optical bandgap increases (blue), decreases (red) and remains unchanged (black), and the corresponding simulated TA spectra.

these two sublattices are connected via a much weaker hydrogen bonding and electrostatic interactions instead of the stronger covalent or ionic bonding, and (2) the overlapping vibrational density of states and the coupling between these two sublattices are relatively weak given that there is a large energy gap between the phonon modes of these two sublattices due to their large mass difference (Supplementary Fig. 5). Consequently, a much longer time will be needed for these two sublattices to reach a quasi-equilibrium state. This slow heat transfer can be further verified from the extended duration of regime ③ with increasing pump fluence (Supplementary Fig. 5).

Lastly, we ascribe regime ④ to the cooling process of the whole heated lattice which will relax to its initial equilibrium state that is associated with a decrease of the optical bandgap. On the other hand, damped oscillations are superimposed on the TA kinetics in both regimes ② and ④ (Fig. 1d). The former is due to energy modulation by coherent optical phonons (COPs) launched by impulsive stimulated Raman scattering (ISRS) via optical deformation potential interaction whereas the latter can be assigned to energy modulation by CLAPs[11]. As demonstrated below, the phonon coherence in regime ② with resonant MIR pump is distinct from that with off-resonant IR pump and that in regime ④ arises from the thermoelastic effect due to the increase of lattice temperature by heat transfer from the thermalized organic cations to the inorganic octahedra, thereby providing further evidence of the organic and inorganic sublattice coupling.

### Fluence- and temperature-dependent pump-probe measurements

To gain deeper insights into this organic and inorganic sublattice coupling in regime ②, we conduct fluence- and temperature-

dependent TA measurements. As shown in Fig. 2a, within the excitation intensity range of interest, $\Delta A$ amplitude of P1 at a delay of 9 ps increases and scales linearly with pump fluence, indicating an enhanced reduction of exciton energy. To quantify this reduction, we first phenomenologically fit the low-temperature linear absorption spectrum using a well-established quantum-well absorption model. Details of the curve-fits (shown in Fig. 1c) and the model description can be found in Supplementary Note 1, Supplementary Table 1, and our previous work[11]. By considering the variation of the exciton energy and linewidth broadening after photoexcitation, we successfully reproduced the shape of transient $\Delta A$, as shown in a representative TA spectrum (inset of Fig. 2b). Details of the fitting can be found in Supplementary Table 2. The estimated $\Delta E_g$ is on the order of 0.1 meV within the excitation range of interest and exhibits a linear dependence on pump fluence (Fig. 2c), suggesting a weak and linear perturbation regime. On the other hand, with increasing temperature, $\Delta A$ amplitude of P1 decreases which is associated with the disappearance of P2 due to thermal heating-induced linewidth broadening effect (Fig. 2d). The decreased $\Delta A$ amplitude of P1 may be due to reduced absorption cross-section of PEA cation arising from enhanced thermal motion of H and I atoms, as well as from the weakened hydrogen bonding because of lattice thermal expansion at elevated temperatures. Temperature-dependent TA kinetics of P1 in the first ~15 ps is shown in Fig. 2e, which displays an initial steady increase followed by a decrease of $\Delta A$ amplitude that is superimposed with damped oscillations contributed by COPs[11]. To quantify this rise time, we fit the TA kinetics of P1 in the first ~15 ps using the equation $y = y_0 + C[1 - \exp(-\frac{t}{\tau_{rise}})]$, where $y_0$ and $C$ are the constants, $\tau_{rise}$ is the intrinsic rise-time which corresponds to the IVER process of PEA cation. Representative curve-fit of P1 kinetics

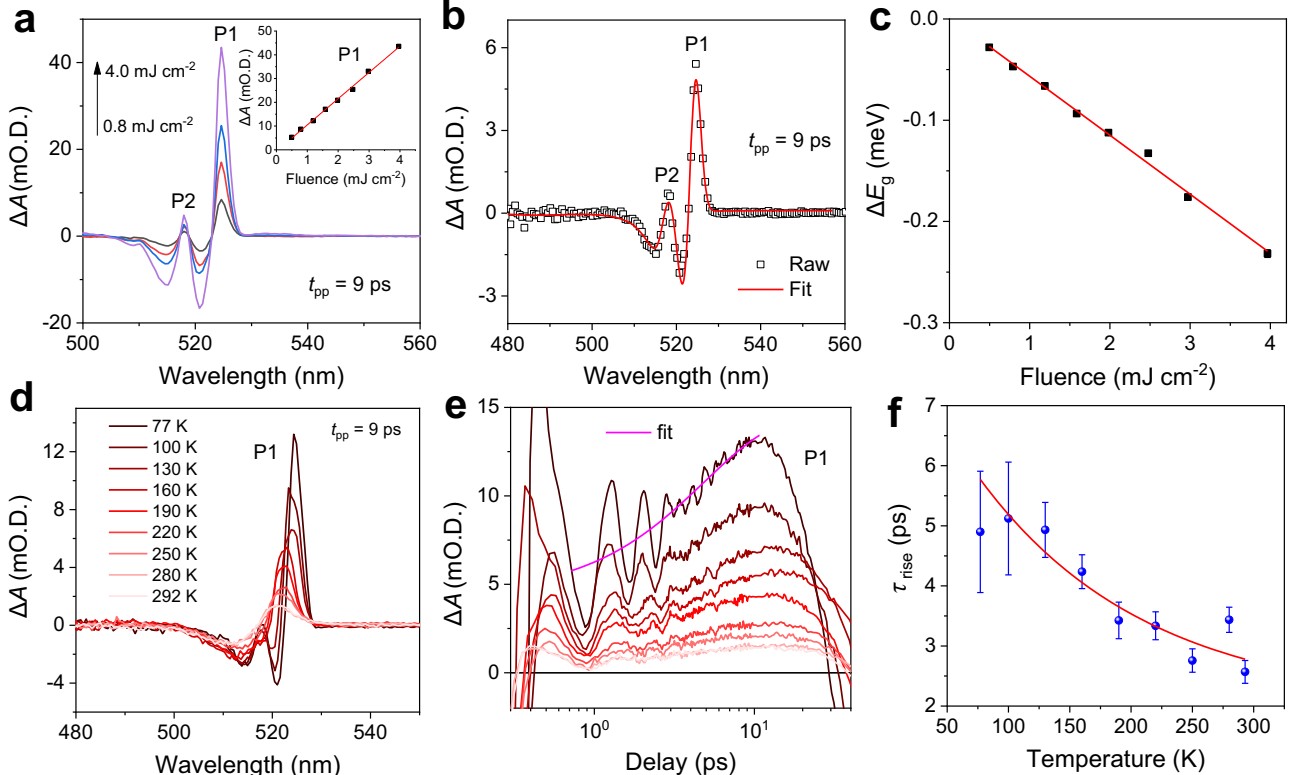

**Fig. 2 | Fluence- and temperature-dependent organic and inorganic sublattice coupling with pump at 3.3 μm in (PEA)₂PbI₄ films. a** Fluence-dependent TA spectrum at a delay of 9 ps at 77 K. Inset: fluence-dependent $\Delta A$ amplitude monitored at P1 (black scatters) and the linear fit (red line). **b** representative TA spectrum (black scatters) at a delay of 9 ps with fluence of 0.5 mJ cm⁻² and the curve fit (red curve). **c** Fluence-dependent bandgap shift for EX state (black scatters) and the linear fit (red line). The error bars are from the curve fit in (**b**). **d** Temperature-dependent TA spectra at a delay of 9 ps with a fluence of 1.7 mJ cm⁻². **e** The corresponding temperature-dependent TA kinetics (from **d**) probed at P1 over the first 15 ps. The magenta line is the curve fit at 77 K. **f** Blue circles: temperature-dependent rise time obtained from fitting the curve in (**e**) (magenta line). The red curve is the exponential-decay fit. The error bars are from the curve-fits in (**e**).

at 77 K is shown in Fig. 2e., which yields a rise-time of $5 \pm 1$ ps that is independent of pump fluence (Supplementary Fig. 6). On the other hand, with increasing temperature, this IVER process becomes faster (Fig. 2f), which is expected given that the phonon-phonon scattering rate is enhanced at elevated temperatures due to the increase of phonon occupation number.

## Sublattice coupling unveiled by optical phonon coherence

To further verify the coupling between the organic PEA and the inorganic PbI₆ sublattice, we examine the fingerprint of optical phonon coherence in the few tens of picoseconds. Coherent phonon spectroscopy allows one to glimpse into the ultrafast lattice dynamics and the coupling between electronic and lattice vibrational degrees of freedom under nonequilibrium conditions. As demonstrated in our previous work[11], the phonon coherence spectrum generated by ultrafast pump-probe measurement involves both coherent phonon generation and detection processes. As for the COP's generation process, the ultrafast dipolar interaction between the phonon mode and electric field of the laser pulse launches COPs via optical deformation potential interactions[11,42]. In general, the optical pumping induced lattice dynamics can be described by the classical driven damped harmonic oscillator model given by:

$$\frac{d^2 Q}{dt^2} + 2\beta \frac{dQ}{dt} + \Omega_0^2 Q = \frac{F}{\mu} \qquad (1)$$

where $Q$ is the lattice displacement, $\beta$ is the damping constant, $\mu$ is the reduced lattice mass, $\Omega_0$ is the natural frequency of the oscillator, and $F$ is the driving force arising from dipolar interactions between the

phonon mode and the electric field of the laser pulse. Within the assumptions of two-band process dominated Raman tensor and vanishing decay rate of the coupled carriers, the driving force for an opaque sample consists of contributions from both virtual and real excitations[11,43]:

$$F(\Omega) = -C\Xi \left[ \frac{\partial \epsilon_R}{\partial \omega} + 2i \frac{\epsilon_{Im}}{\Omega} \right] I(\Omega) \qquad (2)$$

where $C$ is a prefactor, $\epsilon_R$ and $\epsilon_{Im}$ are, respectively, the real and imaginary parts of the dielectric constant, $\Xi$ is the deformation potential constant. And $I(\Omega) = \int_{-\infty}^{+\infty} e^{i\Omega t} |E(t)|^2 dt$, where the electric field of the laser pulse $E = A e^{-t^2 / 2\tau_L^2} \cos \omega_L t$, $A$ is the electric field amplitude, $\tau_L$ is the laser pulse duration, $\omega_L$ is the central pulse frequency. The first term in the brackets of Eq. (2) corresponds to the virtual excitation which is the case for the ground-state ISRS whereas the second term in the brackets of Eq. (2) refers to the real excitation arising from single- or multiple-photon excitation. The solution to Eq. (2) has the expression (Supplementary Note 2)[43]:

$$Q = Q_0 e^{-\beta t} \cos\left( \sqrt{\Omega_0^2 - \beta^2}\, t + \phi \right) \qquad (3)$$

where $Q_0$ is the lattice displacement amplitude that is proportional to the pump intensity, and $\phi$ is the phase constant which is related with the ratio of virtual excitation to that of real excitation. As for the COPs detection process, the induced lattice displacement will modify the linear susceptibility $\chi$ and hence the absorbance of the sample.

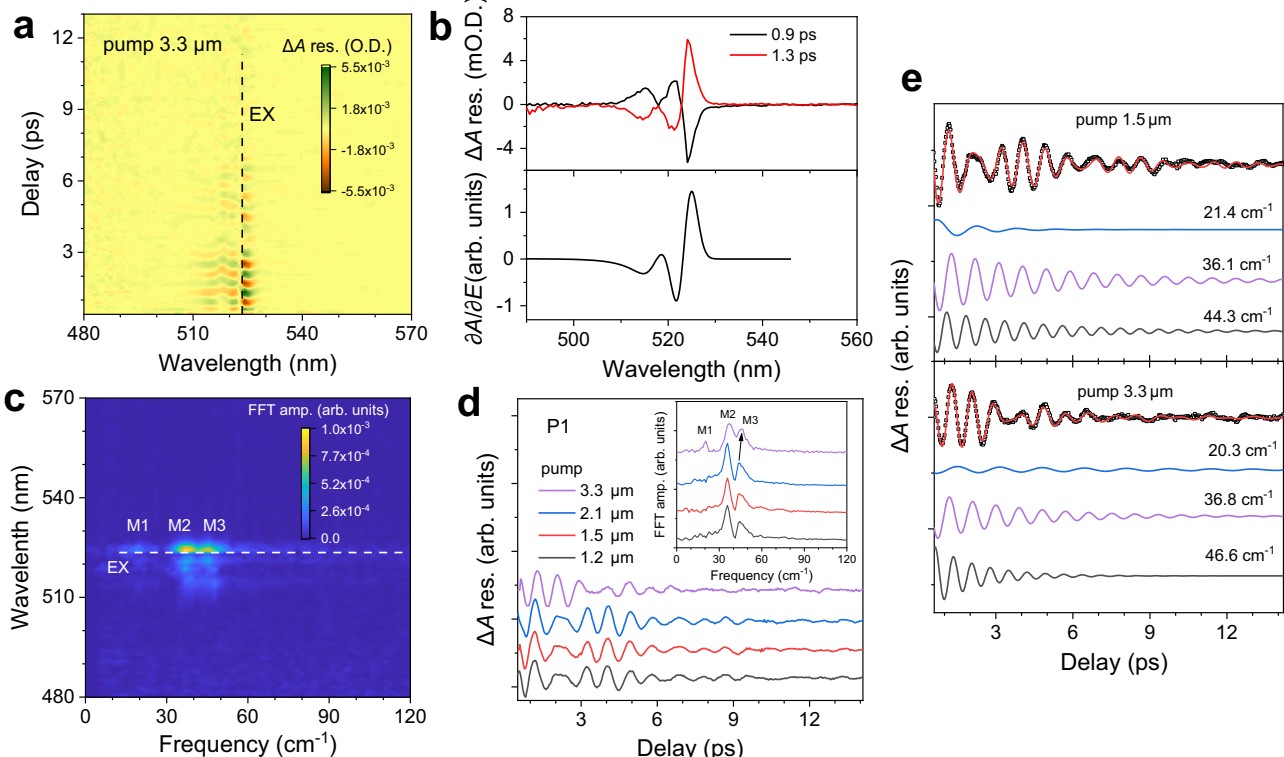

**Fig. 3 | Organic and inorganic sublattice coupling in (PEA)₂PbI₄ revealed by coherent phonon spectroscopy at 77 K. a** 2D contour-plot of time-domain phonon coherence spectrum pumped at 3.3 μm with a fluence of 2 mJ cm⁻². Black dashed line is a guideline to the eye marking the EX peak position. **b** Phonon coherence-induced TA modulation probed at 0.9 and 1.3 ps as compared to spectral profile of the calculated $\partial A/\partial E$. **c** 2D contour-plot of frequency-domain phonon coherence spectrum. White dashed line is the guideline to the eye marking the EX peak position. **d** TA oscillation kinetics probed at P1 for different pump energies. Inset: FFT spectrum of oscillation kinetics for different pump energis. **e** TA oscillation components and the curve-fits for off-resonant and resonant excitations of PEA cation at 1.5 μm and 3.3 μm, respectively. Details of the fits are given in Table 1.

Figure 3a shows the time-domain phonon coherence spectrum for resonant pump (i.e., 3.3 μm) of the PEA cation. The phonon coherence spectrum is obtained by subtracting the non-oscillatory components of the TA kinetics that are fitted using a sum of multiple-exponential decay functions (Supplementary Fig. 7). As expected, the beating map shows a 1π phase shift between the red and blue sides of the EX peak position because of COPs induced energy modulation of the electronic transition[11]. The COPs-induced $\Delta A$ modulation amplitude exhibits a linear dependence on pump fluence regime (Supplementary Fig. 8), indicating a weak and linear perturbation. Meanwhile, the phonon coherence spectrum displays a probe energy-dependent oscillation amplitude (Fig. 3b), which is governed by the COPs detection process. This probe energy dependence can be well explained on the basis of lattice displacement-induced absorbance change because of variation of the linear susceptibility, as given by[11]:

$$\Delta A = \frac{\partial A}{\partial E}\frac{\partial E}{\partial Q}Q = \frac{\partial A}{\partial E}\sum_i \Xi_i Q_i \qquad (4)$$

where $\Xi_i$ is the deformation potential constant for the $i$th phonon mode. As seen in Fig. 3b, the calculated profile of $\frac{\partial A}{\partial E}$ well reproduces the probe energy-dependent modulation.

We then perform a fast Fourier transform (FFT) to the beating map of Fig. 3a to determine the activated optical phonon modes. Figure 3c displays a 2D contour-plot of the calculated FFT amplitude spectrum. As shown, the FFT amplitude spectrum is weakest around the EX peak position with three activated optical phonon modes M1 (~20.6 cm⁻¹), M2 (~37.0 cm⁻¹), and M3 (~45.7 cm⁻¹). These optical phonon modes arising from the twisting or bending motion of Pb−I network mainly possess transverse optical (TO) phonon features[44].

Next, we examine the phonon coherence spectra at P1 obtained using different pump energies. Distinct TA spectra signatures are obtained using resonant (MIR laser pulse at 3.3 μm) vs. off-resonant excitation (e.g., IR laser pulse at 1.5 μm) – see Supplementary Fig. 9 for a side-by-side comparison. Figure 3d shows that when the pump is off-resonant with the N − H stretching mode of PEA cations, the phonon-induced oscillation kinetics and the associated FFT spectra are nearly the same irrespective of pump energy, which is expected for these COPs launched by off-resonant ISRS with the same driving force. However, when the pump is resonant with the N − H stretching mode, the oscillation kinetics and the corresponding FFT spectra (Fig. 3d inset) are different. Compared with off-resonant IR pump, the relative intensities of M1 and M3 modes with respect to that of M2 mode are higher; and the M2 and M3 modes are blue-shifted for resonant MIR pump. To delve into these differences, we deconvolve the oscillation kinetics monitored at P1 for resonant pump of 3.3 μm and off-resonant pump of 1.5 μm using a sum of three damped cosine functions:

$$y(t) = \sum_{i=1}^{3} B_i \cos(\omega_i t + \phi_i)e^{-\frac{t}{\tau_i}} \qquad (5)$$

where $B_i$, $\omega_i$ $\tau_i$, and $\phi_i$ are respectively, the amplitude, frequency, relaxation time, and initial phase of the $i$th oscillation component. The intrinsic phonon frequency $\Omega_i$ is obtained as $\Omega_i = \sqrt{\omega_i^2 + \beta_i^2}$, where $\beta_i = 1/\tau_i$ is the damping rate. Details of the fitting are given in Table 1 and Fig. 3e. As compared to off-resonant pump, we find that under resonant pump: (1) both M2 and M3 modes are blue-shifted whereas M1 mode is red-shifted which suggests the softness of the perovskite lattice; (2) amplitudes of M1 and M3 modes are stronger; (3) relaxation time of M1 mode is longer whereas that of the M2 and M3 modes are

**Table 1 | Fitting details of Fig. 3e using Eq. (5). Here, *c* is the speed of light in the vacuum**

| Pump (μm) | Mode | *B* (mOD) | *τ* (ps) | *k* = $\Omega_0/c$ (cm$^{-1}$) | *φ* (rad) |
|---|---|---|---|---|---|
| 1.5 | M1 | 3.7 ± 0.8 | 1.5 ± 0.3 | 21.7 ± 0.8 | −3.0 ± 0.1 |
| | M2 | 8.2 ± 0.4 | 4.3 ± 0.2 | 36.1 ± 0.1 | −2.4 ± 0.1 |
| | M3 | 5.5 ± 0.3 | 4.1 ± 0.3 | 44.3 ± 0.1 | −2.8 ± 0.1 |
| 3.3 | M1 | 0.7 ± 0.1 | 8.5 ± 0.1 | 20.2 ± 0.1 | 0.5 ± 0.1 |
| | M2 | 4.9 ± 0.3 | 3.4 ± 0.2 | 36.6 ± 0.2 | −1.9 ± 0.1 |
| | M3 | 8.1 ± 0.3 | 1.7 ± 0.1 | 46.6 ± 0.3 | 0.7 ± 0.1 |

shorter; (4) phases of the activated modes are changed. The blue-shift of the phonon modes indicates that the lattice becomes stiffer which is consistent with the distortion of inorganic PbI$_6$ octahedra observed in this regime. On the other hand, these phonon shifts also imply that the proposed reduction of oscillator strength is not likely the origin of regime ② which would not change the frequencies of the activated phonon modes. Furthermore, the change in amplitude, phase and relaxation time of the activated phonon modes indicate that the potential energy surface associated with vibrational anharmonicity of the inorganic sublattice is different. Apart from (PEA)$_2$PbI$_4$, we also observe a decrease in the optical bandgap associated with phonon frequency shifts of PbI$_6$ octahedra after resonant excitation of the organic cation in 2D hexylammonium lead iodide ((HA)$_2$PbI$_4$) perovskite (Supplementary Fig. 10), further demonstrating the organic and inorganic sublattice coupling in hybrid halide perovskites. Nevertheless, compared to (PEA)$_2$PbI$_4$, the phonon frequency shifts in (HA)$_2$PbI$_4$ are smaller, indicating a weaker organic and inorganic sublattice coupling.

Note that the COPs' generation mechanisms for both off-resonant and resonant pumps are ISRS. Nevertheless, the driving force of the lattice displacement for the former is governed by the virtual excitation (i.e., the first term in the brackets of Eq. (2)) whereas for the latter, there is a significant contribution from the real excitation (i.e., the second term in the brackets of Eq. (2)) due to the MIR absorption governed by N−H stretching motion. This nontrivial contribution from the real excitation leads to an enhanced linear response of vibrational amplitude to the pump fluence (Supplementary Fig. 11). The presence of this non-trivial real excitation with resonant pump would also well explain the phase differences of the activated phonon modes under resonant pump compared to those under off-resonant pump. Moreover, if the phonon perturbation is in the linear range, the phonon-induced absorbance modulation amplitude will linearly depend on the pump fluence since the driving force is proportional to the pump power. On the other hand, resonant coupling between the high-order overtone and the 3.3 μm MIR pump pulse (~3030 cm$^{-1}$) is unlikely given that vibrational frequencies of the inorganic sublattice are below ~150 cm$^{-1}$ such that a phonon overtone of more than 20-orders would be required[21,44]. Similarly, electronic excitation from multiple-photon absorption (i.e., needing simultaneous absorption of seven-photons) would also be negligible. Thus, the MIR pump pulse couples to the vibrational motion of both inorganic PbI$_6$ octahedra and organic PEA cation. For the former, it leads to generation of COPs; whereas for the latter, it results in the depletion of MIR absorption which in turn reduces the exciton energy that leads to distinct TA features.

**Presence of coherent longitudinal acoustic phonons**

Next, we investigate the CLAPs-induced oscillation kinetics after phonon down-conversion process (i.e., IVER within PEA cations followed by heat transfer from thermalized PEA cations to PbI$_6$ octahedra) in regime ④. After subtracting the non-oscillatory components which are fitted with multiple exponential decay functions, we obtain the time-domain of CLAPs-induced beating map, as shown in Fig. 4a. Details of

the fitting can be seen in Supplementary Fig. 7. Compared to regime ②, a much longer oscillation period of ~400 ps (~2.5 GHz) corresponding to the acoustic phonon mode is present. This phonon frequency is much smaller than those in most reports[9,45–49] and other conventional semiconductors[50–54], but is comparable to recent reports on stacked lamellar colloidal CdSe nanoplatelet[55] and cobalt supercrystals[56]. The beating map shows a similar probe energy-dependent profile (red curve in Fig. 4a) with that induced by COPs, indicating energy modulation of the electronic transition. Meanwhile, the oscillation period shows a film thickness dependence which is consistent with its long-itudinal vibrational feature (Supplementary Fig. 12). Furthermore, with increasing pump fluence, the CLAPs-induced oscillation is delayed which can be attributed to an even more inefficient heat transfer such that the lattice displacement takes more time to reach its maximum amplitude (Fig. 4b). Although both the phonon down-conversion process-induced Δ*A* modulation amplitude and the CLAPs-induced oscillation amplitude exhibit linear dependences with pump fluence (Supplementary Fig. 13), the oscillation frequency of these launched CLAPs shows a peculiar anharmonicity with pump fluence. As shown in Fig. 4c, the oscillation frequency decreases with increasing pump fluence, which is associated with the appearance of a sideband that may be due to hybridization of the CLAP mode with other high-energy acoustic phonons arising from strong anharmonic phonon-phonon coupling[57]. The softening of the phonon mode was previously reported in highly excited tellurium[57] and bismuth[58] which were attributed to electronic softening of the crystal lattice due to anharmonic coupling. However, this electronic softening mechanism is inapplicable to our case which possesses trivial asymmetric line shapes in the FFT spectra (Supplementary Fig. 14) and has a negligible contribution from the electronic excitation. Here, we attribute the origin of this phonon softening to the enlarged tensile strain arising from enhanced lattice thermal expansion, which can be understood as follows: with increasing pump fluence, more energy absorbed by PEA cation will be transferred to the low-energy phonon modes of PbI$_6$ octahedra via phonon down-conversion process. This then leads to a more heated PbI$_6$ octahedra, resulting in an enhanced thermal expansion of the overall lattice and increase of the optical bandgap, which is associated with the presence of a new non-equilibrium position with a larger initial lattice displacement and a larger tensile strain[59]. On the other hand, the resultant lattice displacement and the plane strain wave (or CLAPs) will bounce back and forth in the films, leading to a periodic modulation of the electronic transition and hence the absorption of the time-delayed probe pulse. Similar CLAPs-induced beating map with the oscillation frequency of several GHz (Supplementary Fig. 15) is also found in resonantly pumped (HA)$_2$PbI$_4$ films. It is also important to highlight that the mechanism of CLAP generation in this work is different from most of the reports in which CLAPs take place after direct light-matter interactions[60,61]. From these reports, three mechanisms are identified: the thermoelastic effect due to fast increase of the lattice temperature[9,60,61], the deformation potential interaction resulting from modification of the electronic transition due to carrier generation[9,61], and the inverse piezoelectric effect owing to a change of the lattice's equilibrium state when the surface built-in electric field is varied after carrier generation[51,61]. Nevertheless, for our case which does not involve any electronic transitions, the CLAPs are launched due to the heat transfer from organic PEA cation to the inorganic PbI$_6$ octahedra, which in turn generates the thermoelastic stress.

To quantify the magnitude of the induced tensile strain, we estimate these strains based on the temperature-dependent optical bandgap which is dominated by the lattice thermal expansion due to anharmonic inter-atomic potentials (Supplementary Figs. 16, 17)[62]. As expected, the induced tensile strain increases with increasing pump fluence where the out-of-plane (i.e., along **c**-axis of the unit cell) strain growing from 0.01% at a pump fluence of 0.3 mJ cm$^{-2}$ to 0.09% at a pump fluence of 2 mJ cm$^{-2}$ (Supplementary Fig. 17). Moreover, the out-

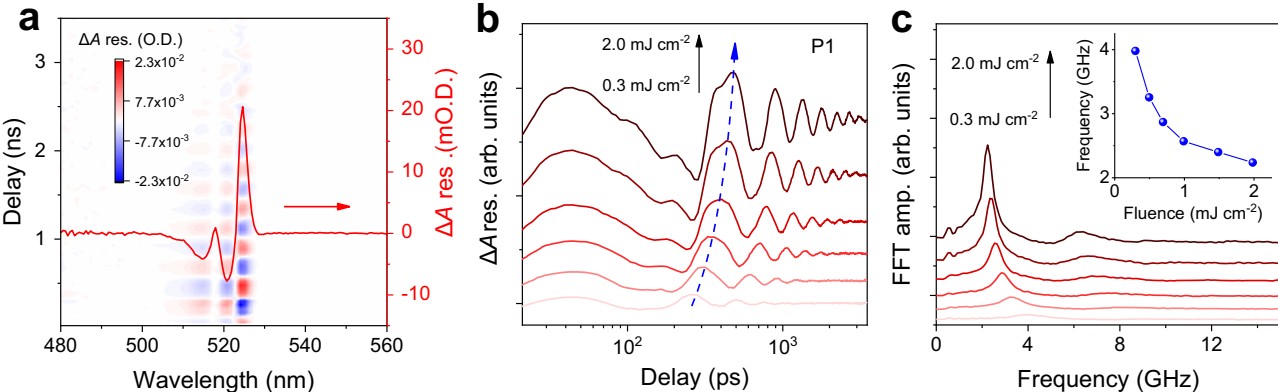

**Fig. 4 | Fluence-dependent CLAP oscillation kinetics. a** 2D contour-plot of the time-domain beating map caused by thermally driven soft CLAPs pumped at 3.3 μm with a fluence of 2 mJ cm$^{-2}$. Red curve is the TA residual at a delay of 240 ps. **b** Fluence-dependent oscillation kinetics monitored at P1. The blue dashed arrow is a guide to the eye showing the delayed oscillation onset with increasing pump fluence. **c** Fluence-dependent FFT spectra of (**b**). Inset shows the oscillation frequency as a function of pump fluence.

of-plane strain is much larger than that of the in-plane strain because of a much weaker van der Waals bonding between the soft organic spacers with a smaller volume occupancy and thus expand/contract more easily compared to the stronger covalent bonding of PbI$_6$ octahedra with a larger volume occupancy.

## Discussion

Coupling between the vibrational motion of the PEA cation and PbI$_6$ octahedra involves both mechanical and electronic couplings[63]. For the former, the nonlinear interaction potential results in off-diagonal elements in the vibrational Hamiltonian matrix which mixes the high-energy phonon modes of PEA cation with the low-energy phonon modes of PbI$_6$ octahedra, creating new pathways for energy relaxation and dissipation. This mechanical coupling agrees well with the observed slow heat transfer process in regime ③. Such weak mechanical coupling which means a small sound velocity, indicates that the cross-plane thermal conductivities of 2D layered halide perovskites are intrinsically low and insensitive to the type of the organic moiety, which agrees well with previous results[12,64]. As for the latter, the interaction between the electronic degree of freedom of the PEA cation and the vibrational motion of the PbI$_6$ octahedra will modify the potential energy surface, and thereby the density of states, vibrational anharmonicity and dispersion of PbI$_6$ octahedra[63]. This electronic coupling is again consistent with the observed modification of the activated phonon modes (i.e., frequency and dephasing time) of PbI$_6$ octahedra in regime ②. On the other hand, this electronic coupling is in line with the observed reduction of optical bandgap, which is likely due to the varied distortion of PbI$_6$ octahedra, as demonstrated below. Similar strong coupling was also reported in which vibrational motion of organic cation leads to screening of excitonic features[65] and reduction of biexciton binding energy[21]. Such strong coupling could offer us an exciting lever to exert ultrafast control over the optoelectronic properties of 2D halide perovskites by simply modulating the local motion of the organic sublattice[66].

To shed further light on the organic and inorganic sublattice coupling in regime ②, we conduct ab initio molecular dynamics (AIMD) simulations to verify the reduced bandgap under the effect of N−H stretching motion. The calculated IR spectrum of PEA cation and the equilibrium optimized crystal structure of (PEA)$_2$PbI$_4$ are shown in Supplementary Fig. 18. To simulate the effect of N−H stretching motion, we employ the same approach reported recently by Gallop et al.[66] via adjusting the atomic velocities along the normal modes corresponding to N−H stretching motion of the PEA cation. Details of the AIMD simulations can be found in the Methods section. As demonstrated in previous reports, the perovskite's bandgap is directly correlated with the distortion index of the inorganic PbI$_6$ octahedral (i.e., $D = \frac{1}{6}\sum_{i=1}^{6}\frac{(|l_i-\bar{l}|)}{\bar{l}}$, where $l_i$ and $\bar{l}$ are the individual Pb−I bond length and the average of six bond lengths, respectively) as well as the in-plane distortion and out-of-plane distortion of the Pb−I−Pb angle[67–69]. We therefore evaluate the bandgap as a function of these parameters. Figure 5a shows a typical simulated bandgap kinetics and distortion index kinetics after exciting the PEA cation. As expected, the bandgap fluctuates with time because of lattice vibrations and anharmonicities. Consistent with the AIMD simulations, the bandgap of (PEA)$_2$PbI$_4$ is reduced with increasing excitation energy of N−H stretching motion. Moreover, the distortion index of the inorganic octahedra exhibits a correlated (increasing or decreasing) trend with the reduced bandgap (Fig. 5b) while other two parameters show no clear correlations (Supplementary Fig. 19). This suggests that the reduced bandgap after exciting N−H stretching motion is more likely because of enhanced distortion index of the inorganic octahedra. Note that the reduced bandgap (~0.1 meV) in our measurement is nearly 2 orders of magnitude smaller compared to our simulated result (~10 meV). This may be due to the smearing effects from the grain boundaries and defects or because we excite all the PEA cations in our simulations while only a small fraction of PEA cations is excited in the sample (Supplementary Note 3). Further studies may be needed to clarify the differences between the experimental results and theoretical calculations. Apart from the reduced bandgap, the electronic coupling between the organic and inorganic sublattices can be further verified from the simulated bandgap kinetics. As shown in Fig. 5c, even in the absence of N−H stretching motion, the FFT spectrum exhibits a non-trivial contribution from the mode at ~3100 cm$^{-1}$, which well matches the mode of the N−H stretching motion. Moreover, this signal is significantly enhanced when directly exciting the N−H stretching mode (Fig. 5c inset). Furthermore, the N−H bond length increases with the growth of excitation energy (Supplementary Fig. 20), which can be attributed to the thermal expansion of organic cation due to IVER.

In summary, we elucidate the interplay between the organic and inorganic sublattices in 2D hybrid halide perovskites through the application of TA spectroscopy and coherent phonon spectroscopy and supported by AIMD simulations. Resonant pump of the PEA cation results in modification of both electronic and lattice degrees of freedom of PbI$_6$ octahedra by changing its structural distortion. The weak coupling between these two sublattices leads to a slow heat transfer process, in which the resultant lattice strain launches coherent longitudinal acoustic phonons. Our work injects fresh insights into the organic and inorganic sublattice coupling that would be important for clarifying

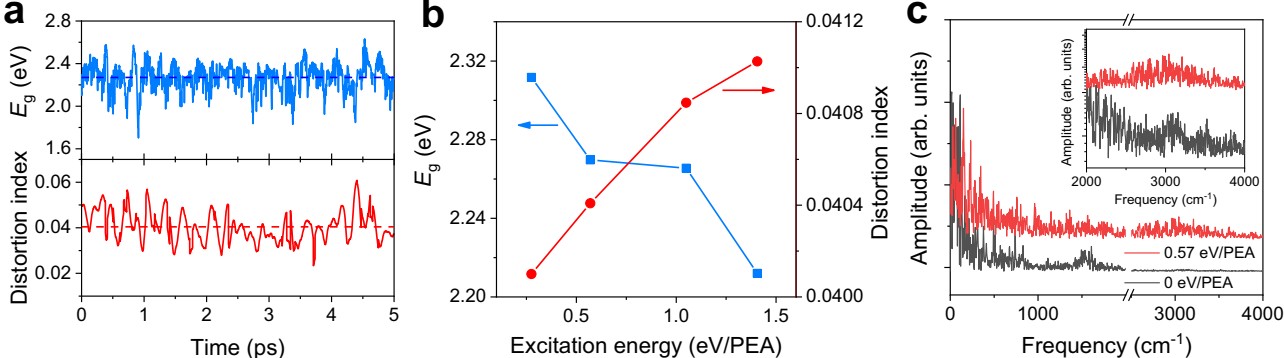

**Fig. 5 | AIMD simulations of direct excitation of PEA cation. a** Typical AIMD simulated bandgap kinetics and distortion index kinetics after exciting N – H stretching motion with an energy of 0.57 eV/PEA. The dashed lines correspond to the average values. **b** Averaged bandgap and distortion index as a function of excitation energy. **c** FFT spectra of the bandgap oscillation kinetics for excitation energy of 0 eV/PEA (black curve) and 0.57 eV/PEA (red curve). Inset is a zoom-in of the spectra.

additional energy relaxation pathways for photoexcited energetic carriers as well as enabling ultrafast control over the optoelectronic properties of 2D halide perovskites through nonlinear phononics[66].

## Methods

### Sample fabrication

$(PEA)_2PbI_4$ and $(HA)_2PbI_4$ films were prepared using a standard spin-coating method. The precursor solution was prepared by dissolving 2:1 molar stoichiometric ratio of PEAI or HAI (Dyesol) and $PbI_2$ (TCI, 99.99%, trace metals basis) in a solution of N,N-dimethylformamide (DMF, Sigma Aldrich anhydrous, ≥ 99%) with molar concentration of 0.12 M. The $(PEA)_2PbI_4$ and $(HA)_2PbI_4$ films were prepared by spin-coating the precursor on $CaF_2$ substrates at 4000 r.p.m. for 30 s followed by annealing at 70 °C for 10 mins. All the preparation was conducted inside a nitrogen filled glove box.

### Sample characterizations

All the samples were first characterized by X-ray diffraction (XRD) measurements. The XRD patterns were collected using a Bruker-AXS D8 Advance X-ray diffractometer equipped with Cu Kα ($\lambda$ = 1.5418 Å) X-ray source. The measurement was performed in $2\theta$ mode between 10° and 50° with step size of 0.02° and the integration time of 1 s per step. The surface morphology and thickness of the films were measured using AFM (Bruker Bioscope Resolve) working in a contact mode. Temperature-dependent absorption measurement was conducted by directing the white light beam generated through a quartz-tungsten-halogen lamp on the films that were kept in a cryostat cooled by liquid nitrogen. The transmitted light was collected by a pair of lenses via an optical fiber coupled by a spectrometer (Acton, Spectra Pro 2500i) and charge coupled device (CCD) (Princeton Instruments, Pixis 400B). For each measurement, transmitted light of a fused silica was collected under the same conditions, that served as a reference.

### Transient absorption spectroscopy

Transient absorption spectra with the IR or MIR pump were collected using a CCD spectrometer operating in a nondegenerate pump–probe configuration. The pump pulses were generated from an optical parametric amplifier (TOPAS-Prime, Light Conversion) pumped by a 1 kHz regenerative amplifier (Coherent EVOLUTION, 800 nm, 35 fs). The generated pump pulses were then chopped at 500 Hz before directing onto the sample. The amplifier was seeded by a mode-locked Ti-sapphire oscillator (Coherent Vitesse, 100 fs, 80 MHz). The white light continuum probe pulses were produced by focusing the 800 nm fs pulses through a 2 mm-sapphire plate. A 750 nm short pass filter was placed before the sample to filter out the 800 nm residue. All the TA measurements were conducted at 77 K except mentioned otherwise.

### Ab initio molecular dynamics simulations

Ab initio molecular dynamics (AIMD) simulations were carried out using the projector-augmented wave (PAW) method as implemented in the Vienna Ab initio Simulation Package (VASP) code[70–72]. The generalized gradient approximation (GGA) together with Perdew-Burke-Ernzerhof (PBE) exchange correlation functional was used. The van der Waals interactions were also included using zero-damping DFT-D3 method of Grimme[73,74]. The atomic positions of $(PEA)_2PbI_4$ crystal structure were fully relaxed until the Hellman-Feynman forces on each atom were less than 0.01 eV/Å. The uniform grid of $4 \times 4 \times 1$ $k$-mesh in the Brillouin zone and the energy cutoff for wave functions of 450 eV were used in the structural optimization.

To understand the effect of exciting PEA cations on the lattice distortions and bandgap change of $(PEA)_2PbI_4$, we first performed equilibrium AIMD simulations at 300 K in the canonical ensemble (NVT) using Nosé-Hoover thermostat with a timestep of 0.5 fs[75,76]. The trajectory was equilibrated for 5 ps and then out of equilibrium AIMD was performed in the microcanonical (NVE) ensemble with a timestep of 1 fs and total simulation time of 5 ps. In the out-of-equilibrium AIMD simulation, the hydrogen atoms from $-NH_3$ groups in the four PEA cations had their velocities increased in the antisymmetric mode, while in the other four PEA cations, the velocities of hydrogen atoms were increased in the symmetric mode. Different velocity increment resulted in the average excitation energies of 0, 275, 510, and 1406 meV per PEA cation. For the AIMD simulations, an energy cutoff of 400 eV was used for the plane wave basis and gamma point sampling was used for the Brillouin zone. The distortion index based on Pb−I bond length is defined as $DI = \frac{1}{n} \sum_{i=1}^{n} \frac{|l_i - l_{av}|}{l_{av}}$, where $l_i$ is the distance from the central Pb atom to the $i^{th}$ coordinating I atom, and $l_{av}$ is the average bond length. AIMD simulations were conducted using the OVITO software[77].

## Data availability

The data underlying this study are openly available in DR-NTU (Data) at https://doi.org/10.21979/N9/CBJGV2.

## Code availability

Custom code is available from the corresponding author upon request.

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

## Acknowledgements

This research/project is supported by the Ministry of Education under its AcRF Tier 2 grants (MOE-T2EP20120-0004 (T.C.S) and MOE-T2EP50120-0013 (T.C.S)); and the National Research Foundation (NRF) Singapore under its NRF Investigatorship (NRF-NRFI2018-04 (T.C.S)) and the Competitive Research Programme (NRF-CRP25-2020-0004 (T.C.S)). J.Y. acknowledges the financial support from the Hong Kong Polytechnic University (grant P0042930 and PolyU 25300823).

## Author contributions

T.C.S. and J.F. conceived the idea and designed the experiments. J.F. performed the spectroscopic characterization and conducted the sample characterization. M.F and W.Y. conducted the FTIR and AFM measurements. T.B., J.Y and Q.X. conducted the DFT and AIMD calculations. J.F. and T.C.S. analysed the data and wrote the manuscript. All authors discussed the results and commented on the manuscript at all stages. T.C.S. led the project.

## Competing interests

The authors declare no competing interests.
