## [Peer Review File · Nature Communications]

Organic and inorganic sublattice coupling in two-dimensional lead halide perovskitesReviewers' comments:

Reviewer #1 (Remarks to the Author):

In their manuscript, Fu et al., probe the electronic and mechanical coupling in the case of prototypical $n=1$ layered perovskite: PEA₂PbI₄. Although the experimental data and analysis is quite interesting, I will argue that the theoretical approach employed is not at the state-of-the-art to justify publication at Nature Communications.

The authors employ a standard DFT methodology to calculate the ground state structure, and then stretch the N-H bond length in an effort to probe the inorganic-organic coupling. This association is assumed to be justified as the N-H is clearly the point-of-contact between the spacer and the octahedra. However, the method misses a series of important underlying mechanisms. First, the dynamical motion of the organic (and inorganic) moieties. It is well-known that the molecules are moving around the ground state position, an effect that will strongly impact the reported values (i.e., for the angles, and the band-gaps). Second, even within a static description of the lattice, polymorphism as defined in the work of Zunger et al., [PRB 101, 155137 (2020)] and the anharmonicity of the lattice at room temperature, would also distort the ground state structure. These two (related) effects are critical for this work, as the values discussed (in Figure 5b) are in the order of just a few meV and less than 0.1 degrees...! Which brings me to my last comment: I sincerely doubt that such precision on a model static description of the structure is reliable, and does not depend on for example the choice of functional for the relaxation, the starting position/rotation of the molecules and so on. Consequently, I do not believe that the calculations provide any real evidence that the out-of-plane angle really correlates with the E_g change.

Moreover, there are a couple of sentences in the introduction that do not make sense to me. First, in line 45 the authors claim that the materials are held together by van der Waals forces. Yet, the organic spacers are in fact cations (PEA), hence there is clearly an ionic forces that significantly contribute to the stability of the materials. Second, in line 52 they say that the organic cations with varying lengths can alter the thickness of the inorganic layers. Most likely, the authors mean the interlayer distance between the inorganic layers?

To conclude, I cannot recommend publication of the work with this level of calculations claiming a validation of the experimental work.

Reviewer #2 (Remarks to the Author):

Fu et al present a very timely and systematic investigation of the lattice dynamics in two-dimensional perovskites. By using mid-IR pump and visible probe, they probe the role of organic-inorganic interactions in the exciton energetics and optical and acoustic phonon dynamics in the inorganic sublattice. The results present a very interesting scenario, where excitation of local vibrations in the organic cation seems to produce dynamic blue and red shifts in the exciton energy. But I have a few critical concerns about the presented interpretations and the mechanisms behind them:

1. The authors should discuss the “phonon thermalization” in the organic cation. Given that these are localized vibrations within the organic cation, a more suitable nomenclature would be intramolecular energy redistribution. Typically, one needs a highly excited molecule (in the excited state) to access the higher energy vibrational quanta, which enforces a

redistribution of the vibrational energy over the entire molecule. I am specifically referring to a series of experimental and theoretical works on this topic, such as: J. Chem. Phys. 82, 2975–2993 (1985), J. Phys. Chem. 1996, 100, 31, 12735–12756. Some discussion of this process and possible timescales involved in reference to the previous literature is needed to elucidate on this hypothesis. A mid-IR pump-mid IR probe experiment will provide unambiguous probe of these dynamics, although I understand that such an experiment may require additional instrument capabilities and thus may be out of the scope of this manuscript.

2. The DFT calculation presented at the end specifically considers an increase in the N-H bond-length to reproduce the strain on the inorganic lattice. Why does the intramolecular vibrational relaxation result in the increase in the N-H bond length? As noted earlier, vibrational energy redistribution should happen over the entire molecule, and it may not be trivial to predict the trajectory of such a redistribution. Can DFT calculations on the molecule alone provide more insights into it?

3. The mechanism of downconversion of the organic cation vibrations into long-range phonons in the inorganic lattice is also unclear. While the organic vibrations are localized (and not phonons) and have no well-defined momentum, the phonons in the lattice have. How is momentum conservation not violated in such a process? Moreover, the authors observe the generation of coherent strain waves. What is the suggested mechanism through which the relatively independent motion of organic moieties leads to such a long-range coherence? Even if one assumes that the MIR pump imposes coherence between the motion of the organic cation, can such a coherence last for several tens of picoseconds to enable the generation of coherent acoustic phonons?

Reviewer #3 (Remarks to the Author):

The authors report on a study of the 2d hybrid perovskite (PEA)₂PbI₄ under mid-infrared resonant excitation of the organic cations, probing energy transfer between the organic and inorganic lattice. As they note a prior Nature Comm paper by Guo (reference 30) has carried out a similar experiment in the 3d halide perovskites, observing only slow coupling via thermal effects. Here a more complex dynamical response is observed spanning a wide range of time-scales. I think the results are interesting and will be of interest to the broader community interested in the perovskites although I have a number of questions and comments:

1. How is it known that the 3.3 μm pump only couples initially to the organic lattice? Are there higher order phonon resonances from the inorganic lattice that could lie there? How about an impulsive stimulated raman scattering response arising from the sub-gap excitation of the inorganic lattice? This would give rise to the linear in fluence response and could represent a more direct mechanism for generating the observed phonon responses.

2. The very fast response seen in regime 1 is attributed to an optical stark effect e.g. driven directly by the light field. If such a coupling exists and modulates the bands associated with the inorganic lattice, does this not then directly couple to the inorganic lattice through e.g. deformation potential coupling? This would provide another mechanism that should be discussed in the answer to question 1 above.

3. The slower response in regime 3 as I understand it is arising from a pure heating of the

inorganic lattice which leads to lattice expansion mediated by acoustic phonons. Presumably the time-scale observed here is just the film thickness divided by sound velocity? It is stated that "It is important to highlight that the mechanism of CLAP generation in this work is different from

previously reported mechanisms of thermoelastic effect, deformation potential interaction, or inverse piezoelectric effect". Is not this response exactly a thermoelastic effect? As soon as coupling occurs to the inorganic lattice it heats and this follows an acoustic response as originally investigated by e.g. Thomsen et al. decades ago (Phys. Rev. B, 34, 4129 (1986)). Are the authors claiming the mechanism for the CLAP response is distinct from this?

4. Overall the big picture on why these results are important is lacking in the current version. The paper concludes with some comments about how this would impact applications with respect to next generation nonlinear phononics which is unclear. More importantly, it is unclear how the results of this study elucidate our understanding of the optoelectronic functionality of these materials. In the original Guo paper the story was somehow that all of the unique properties of MAPI arise from the inorganic lattice, given the weak coupling. Here there is a strong coupling - so what does this tell us about the key unknown aspects of the 2d perovskites? Without a good answer to this this paper is of interest to specialists in the field but does not truly impact our understanding of these materials.

- Reviewers' comments are in brown
- Our responses are in black
- The sentences revised or added in the manuscript are highlighted in blue
- The sentences that are deleted are strikethrough in red

Reviewer #1:

General comments: In their manuscript, Fu et al., probe the electronic and mechanical coupling in the case of prototypical n=1 layered perovskite: PEA₂PbI₄. Although the experimental data and analysis is quite interesting, I will argue that the theoretical approach employed is not at the state-of-the-art to justify publication at Nature Communications.

Response: We greatly appreciate the reviewer's valuable time in reviewing our manuscript as well as his/her critical comments, which are very helpful for improving our manuscript. We have carefully considered the referee's comments, conducted additional *ab initio* molecular dynamics (AIMD) to support our experimental results. With this revised version of our manuscript, we are confident that we have addressed all the concerns raised by the reviewer and thus our manuscript meets the high-standard quality of *Nature Communications*. Below are our replies to the reviewer's detailed comments.

Comment 1) The authors employ a standard DFT methodology to calculate the ground state structure, and then stretch the N-H bond length in an effort to probe the inorganic-organic coupling. This association is assumed to be justified as the N-H is clearly the point-of-contact between the spacer and the octahedra. However, the method misses a series of important underlying mechanisms. First, the dynamical motion of the organic (and inorganic) moieties. It is well-known that the molecules are moving around the ground-state position, an effect that will strongly impact the reported values (i.e., for the angles, and the band-gaps). Second, even within a static description of the lattice, polymorphism as defined in the work of Zunger et al., [PRB 101, 155137 (2020)] and the anharmonicity of the lattice at room temperature, would also distort the ground state structure. These two (related) effects are critical for this work, as the values discussed (in Figure 5b) are in the order of just a few meV and less than 0.1 degrees...! Which brings me to my last comment: I sincerely doubt that such precision on a model static description of the structure is reliable and does not depend on for example the choice of functional for the relaxation, the starting position/rotation of the molecules and so on. Consequently, I do not believe that the calculations provide any real evidence that the out-of-plane angle really correlates with the Eg change.

Response: We thank the reviewer for the constructive feedback. We agree with the reviewer that the static model may not be reliable to explain the experimental data. To provide further evidence of the reduced bandgap after exciting N-H stretching motion, we conducted additional *ab initio* molecular dynamics (AIMD) simulations. Here, we employ the same approach reported recently by Gallop *et al.* (*Nat. Mater.*, **23**, 88 (2024)) via adjusting the atomic velocities along the normal modes corresponding to N-H stretching motion of the PEA cation. Details of these simulations can be found in the Methods section of main text. **Fig. R1a** shows a typical simulated bandgap kinetics after exciting N-H stretching motion of the PEA cation. As expected, the bandgap fluctuates with time because of lattice vibrations and anharmonicities. Meanwhile, consistent with the AIMD simulations, the bandgap of (PEA)₂PbI₄ is reduced with increasing excitation energy of N-H stretching motion (**Fig. R1d**). We consider several factors that have been widely reported to play a significant role in the bandgap of halide perovskites, such as the distortion index of the inorganic octahedra ($D = \frac{1}{6} \sum_{i=1}^6 \frac{(|l_i - \bar{l}|)}{\bar{l}}$, where l_i and \bar{l} are the individual Pb-I bond length and the average of six bond lengths, respectively), in-plane and out-of-plane distortion of I-Pb-I angle (*Inorg.*

Chem., **44**, 4699 (2005), *Nature*, **620**, 328 (2023)). We simulated their kinetics after exciting the N–H stretching motion and obtained their average values (**Fig. R1**). As shown, only the distortion index of the inorganic octahedra exhibits a correlated (increasing or decreasing) trend with respect to excitation energy, suggesting that the reduced bandgap is more likely because of enhanced distortion index of the inorganic octahedra.

Fig. R1 **a** AIMD simulated bandgap kinetics (top panel) versus distortion index kinetics (bottom panel). **b** AIMD simulated bandgap kinetics (top panel) versus in-plane distortion of the Pb–I–Pb angle (bottom panel). **c** AIMD simulated bandgap kinetics (top panel) versus out-of-plane distortion of the Pb–I–Pb angle (bottom panel). The dashed lines in **a**, **b** and **c** correspond to the average values. The excitation energy of N–H stretching motion is 0.57 eV/PEA. **d** The average bandgap and distortion index as a function of excitation energy. **e** The average bandgap and in-plane distortion as a function of excitation energy. **f** The average bandgap and out-of-plane distortion as a function of excitation energy.

To further verify the organic and inorganic sublattice coupling, we also calculated the FFT spectrum of the simulated bandgap kinetics. As shown in **Fig. R2**, even in the absence of N–H stretching motion, the FFT spectrum exhibits a non-trivial contribution from mode at $\sim 3100\text{ cm}^{-1}$, which well matches the N–H stretching motion. Moreover, this signal amplitude is enhanced significantly when N–H stretching mode is activated, as depicted in **Fig. R2** inset.

Fig. R2 FFT spectra of the bandgap oscillation kinetics for after exciting N–H stretching motion with an excitation energy of 0 eV/PEA (black curve) and 0.57 eV/PEA (red curve). Inset is the zoom-in of the spectra.

As pointed out by the reviewer, our simulated result (~ 10 meV) is nearly 2 orders of magnitude larger compared to the experimental result (~ 0.1 meV). This is likely because we excite all the PEA cations in our simulation while only a small fraction of PEA cations is excited in the sample. To verify this, we first estimate the number of excited PEA in the sample using:

$$n = \frac{P\alpha}{h\nu} \quad (1)$$

where P is the pump power, α is the IR absorption coefficient, and $h\nu$ is the pump energy. For typical pump power of ~ 1 mJ cm $^{-3}$, α of $\sim 2 \times 10^3$ cm $^{-3}$, n is estimated as 3.3×10^{20} cm $^{-3}$.

The number of PEA cations N in 1 cm 3 is calculated using:

$$N = 8 \frac{\rho}{M} N_A \quad (2)$$

where ρ is the mass density (~ 2.55 g cm $^{-3}$), M is the molar mass density (959.19 kg mol $^{-1}$), N_A is Avogadro number. Here, N is estimated as $\sim 1.3 \times 10^{22}$. The fraction of excited PEA cations is then estimated as $n/N \sim 2.6 \times 10^{-2}$. Assuming a linear relationship between the optical bandgap and pump power, which holds for our case based on our experimental results (Figure 2c of the main text), the estimated reduced optical bandgap is thus reasonable.

In view of these feedback, we have cautioned our conclusions and made necessary changes to Figure 5 as well as the main text, accordingly, as shown below.

Figure 5 (a) Typical AIMD simulated bandgap kinetics (top panel) and distortion index kinetics (bottom panel) after exciting N–H stretching motion with an energy of 0.57 eV/PEA. The dashed lines correspond to the average values. (b) Averaged bandgap and distortion index as a function of excitation energy. (c) FFT spectra of the bandgap oscillation kinetics for excitation energy of 0 eV/PEA (black curve) and 0.57 eV/PEA (red curve). Inset is a zoom-in of the spectra.

At line 28 on page 2 of the Abstract:

Here, we examine the coherent dynamics between these sublattices in 2D perovskites using femtosecond pump-probe spectroscopy and *ab initio* molecular dynamics (AIMD) simulations. Unlike off-resonant pumping, resonant excitation of the organic cation sublattice using a mid-infrared femtosecond laser pulse modifies both the electronic transitions and lattice degrees of freedom in the inorganic octahedral sublattice, indicating the existence of electronic coupling. AIMD simulations verify that the reduced bandgap is likely due to the enhanced distortion index of the inorganic octahedra.

At line 401 on page 21 of the main text:

To shed further insights into the organic and inorganic sublattice coupling in regime ②, we conduct AIMD simulations to verify the reduced bandgap under the effect of N–H stretching motion. The calculated IR spectrum of PEA cation and the equilibrium optimized crystal structure of (PEA)₂PbI₄ are shown in Supplementary Fig. S18. To simulate the effect of N–H stretching motion, we employ the same approach reported recently by Gallop *et al.*⁶⁴ via adjusting the atomic velocities along the normal modes corresponding to N–H stretching motion of the PEA cation. Details of the AIMD simulations can be found in the Methods section. As demonstrated in previous reports, the perovskite’s bandgap is directly correlated with the distortion index of the inorganic PbI₆ octahedral (*i.e.*, $D = \frac{1}{6} \sum_{i=1}^6 \frac{(l_i - \bar{l})}{\bar{l}}$, where l_i and \bar{l} are the individual Pb–I bond length and the average of six bond lengths, respectively) as well as the in-plane distortion and out-of-plane distortion of the Pb–I–Pb angle.⁶⁵⁻⁶⁷ We therefore evaluate the bandgap as a function of these parameters. **Error! Reference source not found.**a shows a typical simulated bandgap kinetics and distortion index kinetics after exciting the PEA cation. As expected, the bandgap fluctuates with time because of lattice vibrations and anharmonicities. Consistent with the AIMD simulations, the bandgap of (PEA)₂PbI₄ is reduced with increasing excitation energy of N–H stretching motion. Moreover, the distortion index of the inorganic octahedra exhibits a correlated (increasing or decreasing) trend with the reduced bandgap (**Error! Reference source not found.**b) while other two parameters show no clear correlations (Supplementary Fig. S19). This suggests that the reduced bandgap after exciting N–H stretching motion is more likely because of enhanced distortion index of the inorganic octahedra. Note that the reduced bandgap (~0.1 meV) in our measurement is nearly 2 orders of magnitude smaller compared to our simulated result (~10 meV). This is likely due to the smearing effects

from the grain boundaries and defects or because we excite all the PEA cation in our simulations while only a small fraction of PEA cation is excited in the sample (Supplementary Note S3). Apart from the reduced optical bandgap, the electronic coupling between organic and inorganic sublattice can be further verified from the simulated bandgap kinetics. As shown in **Error! Reference source not found.c**, even in the absence of N–H stretching motion, the FFT spectrum exhibits a non-trivial contribution from the mode at $\sim 3100\text{ cm}^{-1}$, which well matches the mode of the N–H stretching motion. Moreover, this signal is significantly enhanced when directly exciting N–H stretching mode (**Error! Reference source not found.c** inset). Furthermore, the N–H bond length increases with the growth of excitation energy (Supplementary Fig. S20), which can be attributed to the thermal expansion of organic cation due to IVER.

At line 418 on page 21 of the main text:

In summary, we clarify the interactions between the organic and inorganic sublattices in 2D hybrid halide perovskites by using TA spectroscopy, coherent phonon spectroscopy together with AIMD simulations.

At line 462 on page 23 of the main text:

AIMD simulations

The AIMD simulations were carried out using the projector-augmented wave (PAW) method as implemented in the Vienna Ab initio Simulation Package (VASP) code.⁶⁸⁻⁷⁰ The generalized gradient approximation (GGA) together with Perdew-Burke-Ernzerhof (PBE) exchange correlation functional was used. The van der Waals interactions were also included using zero-damping DFT-D3 method of Grimme.^{71,72} The atomic positions of (PEA)₂PbI₄ crystal structure were fully relaxed until the Hellman-Feynman forces on each atom were less than 0.01 eV/Å. The uniform grid of $4\times 4\times 1$ k -mesh in the Brillouin zone and the energy cutoff for wave functions of 450 eV were used in the structural optimizations.

To understand the effect of exciting PEA cations on the lattice distortions and bandgap change of (PEA)₂PbI₄, we first performed equilibrium AIMD simulations at 300 K in the canonical ensemble (NVT) using Nosé-Hoover thermostat with a timestep of 0.5 fs.^{73,74} The trajectory was equilibrated for 5 ps and then out of equilibrium AIMD was performed in the microcanonical (NVE) ensemble with a timestep of 1 fs and total simulation time of 5 ps. In the out-of-equilibrium AIMD simulation, the hydrogen atoms from –NH₃ groups in the four PEA cations had their velocities increased in the antisymmetric mode, while in the other four PEA cations, the velocities of hydrogen atoms were increased in the symmetric mode. Different velocity increment resulted in the average excitation energies of 0, 275, 510, and 1406 meV per PEA cation. For the AIMD simulations, an energy cutoff of 400 eV was used for the plane wave basis and gamma point sampling was used for the Brillouin zone. The distortion index based on Pb–I bond length is defined as $DI = \frac{1}{n} \sum_{i=1}^n \frac{|l_i - l_{av}|}{l_{av}}$, where l_i is the distance from the central Pb atom to the i^{th} coordinating I atom, and l_{av} is the average bond length. AIMD simulations were conducted using the OVITO software.⁷⁵

Furthermore, we have also modified Supplementary Fig. S18, added Supplementary Fig. S19 and Note S3 to SI, as shown below:

Fig. S181 **a** Optimized crystal structure of $(\text{PEA})_2\text{PbI}_4$. **b** Calculated IR spectra of PEA^+ molecule at B3LYP/6-31G(d,p) level (scaling factor = 0.961).

Fig. S19 **a** Schematic of the out-of-plane distortion θ_{out} and in-plane distortion θ_{in} of the inorganic octahedra. The plane is defined by three adjacent Pb atoms that are parallel to the inorganic layer. The AIMD simulated kinetics for the bandgap E_g , **b**, distortion index **c**, θ_{in} **d** and θ_{out} **e** after exciting N–H stretching motion with several excitation energies. **f** Average bandgap E_g , θ_{in} and θ_{out} as a function of excitation energy.

Supplementary Note S3

We note that our simulated result (~ 10 meV) is nearly 2 orders of magnitude larger compared to the experimental results (~ 0.1 meV). This is likely because we excite all the PEA cation in our simulation while only a small fraction of PEA cation is excited in the sample. To verify this, we first estimate the number of excited PEA in the sample using:

$$n = \frac{P\alpha}{h\nu} \quad (11)$$

where P is the pump power, α is the IR absorption coefficient, and $h\nu$ is the pump energy. For typical pump power of $\sim 1 \text{ mJ cm}^{-3}$, α of $\sim 2 \times 10^3 \text{ cm}^{-3}$, n is estimated as $3.3 \times 10^{20} \text{ cm}^{-3}$.

And the number of PEA cation N in 1 cm^3 is calculated using:

$$N = 8 \frac{\rho}{M} N_A \quad (12)$$

where ρ is the mass density ($\sim 2.55 \text{ g cm}^{-3}$), M is the molar mass density ($959.19 \text{ kg mol}^{-1}$), N_A is the Avogadro's number. Here, N is estimated as $\sim 1.3 \times 10^{22}$. The fraction of excited PEA cation is then estimated as $n/N \sim 2.6 \times 10^{-2}$. Assuming a linear relationship between the optical bandgap and pump power which holds for our case based on our experimental results (Figure 2c of the main text), the estimated reduced optical bandgap is thus reasonable.

Comment 2) Moreover, there are a couple of sentences in the introduction that do not make sense to me. First, in line 45 the authors claim that the materials are hold together by van der Waals forces. Yet, the organic spacers are in fact cations (PEA), hence there is clearly an ionic force that significantly contribute to the stability of the materials.

Response: We thank the reviewer for raising this comment. Both the ionic forces and van der Waals forces are present between the organic cation layer in the layered 2D lead halide perovskite A_2PbX_6 . However, only the former contributes to the stability of the structure, *i.e.*, connecting the nearby isolated units A_2PbX_6 together. This can be understood based on **Fig. R3**. To simplify the analysis, we can treat the layered 2D structure as a linear atomic lattice chain. As shown, it is the attractive van der Waals forces rather than the repulsive ionic forces in between A cations that holds the isolated units together. The role of these van der Waals forces is well-recognized in layered 2D Ruddlesden-Popper perovskites (*J. Phys. Chem. Lett.*, 9, 3416–3424 (2018), *Angew. Chem. Int. Ed.*, 61, e202112022 (2022)), which has also been reported in other 2D van der Waals heterostructures (*Nature* 499, 419–425 (2013)) and MXenes (*Adv. Mater.* 26, 992–1005 (2014)). Hence, we respectfully disagree that the ionic forces in between PEA cations lead to the stability of the halide perovskites.

Fig. R3 Schematic of crystal structure of layered 2D lead halide perovskite A_2PbX_4 . The blue and orange dashed lines represent the hydrogen bonding and van der Waals bonding, respectively. The layered 2D structure can be simplified into a linear atomic chain structure with green and dark spheres representing A cation and PbX_6 octahedron, respectively.

In view of this feedback, we have modified the statement at line 46 on page 3 of main text, as shown below:

These octahedra are sandwiched by bilayers of bulkier organic spacers, which are held together by van der Waals forces (*J. Phys. Chem. Lett.*, **9**, 3416–3424 (2018), *Angew. Chem. Int. Ed.*, **61**, e202112022 (2022)) similar to 2D van der Waals heterostructures (*Nature* **499**, 419–425 (2013)) and MXenes (*Adv. Mater.* **26**, 992–1005 (2014)).

Comment 3) Second, in line 52 they say that the organic cations with varying lengths can alter the thickness of the inorganic layers. Most likely, the authors mean the interlayer distance between the inorganic layers?

Response: We thank the reviewer for picking out this misleading description. We have revised this statement at line 52 on page 3 of the main text, as shown below:

Meanwhile, organic cations with varying lengths can alter the interlayer distance between the inorganic layer, thereby modifying both the quantum confinement and the oscillator strength.¹³

Reviewer #2:

General comments: Fu et al present a very timely and systematic investigation of the lattice dynamics in two-dimensional perovskites. By using mid-IR pump and visible probe, they probe the role of organic-inorganic interactions in the exciton energetics and optical and acoustic phonon dynamics in the inorganic sublattice. The results present a very interesting scenario, where excitation of local vibrations in the organic cation seems to produce dynamic blue and red shifts in the exciton energy. But I have a few critical concerns about the presented interpretations and the mechanisms behind them:

Response: We are delighted that reviewer 2 shares the enthusiasm of our findings. We greatly appreciate the reviewer for his/her valuable time for reviewing our manuscript and as well as his/her critical comments which are very helpful for improving our manuscript.

Comment 1) The authors should discuss the “phonon thermalization” in the organic cation. Given that these are localized vibrations within the organic cation, a more suitable nomenclature would be intramolecular energy redistribution. Typically, one needs a highly excited molecule (in the excited state) to access the higher energy vibrational quanta, which enforces a redistribution of the vibrational energy over the entire molecule. I am specifically referring to a series of experimental and theoretical works on this topic, such as: *J. Chem. Phys.* 82, 2975–2993 (1985), *J. Phys. Chem.* 1996, 100, 31, 12735–12756. Some discussion of this process and possible timescales involved in reference to the previous literature is needed to elucidate on this hypothesis. A mid-IR pump-mid IR probe experiment will provide unambiguous probe of these dynamics, although I understand that such an experiment may require additional instrument capabilities and thus may be out of the scope of this manuscript.

Response: We thank the reviewer for raising this constructive feedback that will help strengthen our manuscript. We agree with the reviewer that intramolecular vibrational energy redistribution (IVER) is more suitable for describing the energy cascade process in regime ② in which all the intermediate phonon modes (such as C–H stretching mode, N–H bending mode, C–H bending mode, C–N stretching and bending modes, *etc.*) starting from the excited highest-lying mode (*i.e.*, N–H stretching mode) to the lowest-lying mode will be populated. The duration of this IVER process is normally determined by the vibrational energy transfer (VET) timescale (*Phys. Chem. Chem. Phys.*, 4, 271-278 (2002)) provided that such VET process is dominated by sequential low-order process via intermediate mode. In general, the VET process of high-energy mode in molecules with more than five atoms is in general very fast, leading to fast relaxation of populated vibrations on the time scale of few picoseconds (*Proc. Natl. Acad. Sci.*, 104, 14209 (2007)). This VET process which features the rise time in the populated mode can be usually detected by the time-resolved vibrational spectroscopy techniques, such as IR-pump/IR-probe and IR pump/coherent anti-Stokes Raman-probe. Normally, this VET process which occurs on the timescales ranging from sub-picosecond to several picoseconds depends on the coupling strength and spacing between the parent mode and the daughter mode and is faster for higher-energy daughter modes (*Phys. Chem. Chem. Phys.*, 4, 271-278 (2002), *J. Phys. Chem. A*, 121, 4948 (2017)). For instance, typical VET time of less than one picosecond was found after exciting C–H stretching mode to the intermediate daughter mode in benzene (*J. Phys. Chem. A*, 121, 4948 (2017)) whereas several picoseconds was reported after exciting the highest-lying stretching mode to the lowest-lying daughter modes in CH₂I₂ (*Phys. Chem. Chem. Phys.*, 4, 271 (2002)). Nevertheless, limited by our setup, we are not able to pump the high-energy mode and simultaneously detect the low-energy mode (*i.e.*, two-color pump-probe measurements). More importantly, such VET process in hybrid halide perovskite system has not yet been directly explored experimentally using two-color pump-probe measurements, which requires further studies. Here, we propose that the observed TA feature in the first ~10 ps of regime ② is likely due to the IVER process for the following three reasons: (1) the vibrational population lifetime of excited N–H stretching mode (~3.9 ps, **Fig. R4**) is shorter compared to the duration of regime ②; (2) the timescale of this regime matches well with previously reported IVER process in 2D halide perovskites (*Phys. Rev. Lett.*, 129, 177401 (2022)) as well as molecule CH₂I₂

(*Phys. Chem. Chem. Phys.*, **4**, 271 (2002)) and CH₃I (*J. Chem. Phys.*, **120**, 6973 (2004)) measured using MIR pump-electronic probe spectroscopy.

Fig. R4 Vibrational population dynamics of N–H stretching motion. Black scatters: raw data. Red curve: mono-exponential decay fitting which yields a lifetime of ~3.9 ps.

In view of this feedback, we have modified the statement at line 134 on page 7 of the main text, as displayed below:

We attribute regime ② to the intramolecular **vibrational energy redistribution (IVER)** of PEA cation in which **sequential energy down-conversion takes place through all the intermediate phonon modes starting from the excited N–H stretching mode to the lowest-lying phonon modes**. The timescale of this process normally involves all the intermediate modes and well matches previous MIR pump-visible probe measurements in 2D halide perovskites (*Phys. Rev. Lett.*, **129**, 177401 (2022)) as well as the CH₂I₂ and CH₃I molecules (*Phys. Chem. Chem. Phys.*, **4**, 271 (2002), *J. Chem. Phys.*, **120**, 6973 (2004)).

Comment 2) The DFT calculation presented at the end specifically considers an increase in the N-H bond-length to reproduce the strain on the inorganic lattice. Why does the intramolecular vibrational relaxation result in the increase in the N-H bond length? As noted earlier, vibrational energy redistribution should happen over the entire molecule, and it may not be trivial to predict the trajectory of such a redistribution. Can DFT calculations on the molecule alone provide more insights into it?

Response: We thank the reviewer for this critical feedback. In fact, the intramolecular vibrational energy redistribution process would lead to population of all the vibration modes of the organic cation. In the end, the whole organic cation will be heated up and expanded, which will increase the average bond length of the organic cation, including the N–H bond. To verify this effect, we have employed *ab initio* molecular dynamics (AIMD) simulations and obtained the evolution of the average length of all the N–H bond after exciting N-H stretching motion. As shown in Fig. R5, the N–H bond length fluctuates because of lattice vibrations and anharmonicities. Meanwhile, the fluctuation amplitude is reduced significantly within the first 0.5 ps and is larger for larger excitation energy. Furthermore, the average N–H bond length (at quasi-equilibrium state at intramolecular vibrational redistribution of PEA cation) increases with the growth of excitation energy (Fig. R5 inset).

Fig. R5 Simulated ensemble averaged N-H bond length kinetics after exciting N–H stretching motion with different excitation energies. The inset is the time-averaged N-H bond length as a function of excitation energy.

In view of this concern, we have added Supplementary Fig. S20 to SI and made necessary changes to the main text, as shown below:

Fig. S2 Simulated ensemble averaged N–H bond length kinetics after exciting N–H stretching motion with different excitation energies. Inset is the time-averaged N–H bond length as a function of excitation energy.

And at line 414 on page 21 of the main text,

Apart from the reduced bandgap, the electronic coupling between the organic and inorganic sublattices can be further verified from the simulated bandgap kinetics. As shown in Fig. 5c, even in the absence of N–H stretching motion, the FFT spectrum exhibits a non-trivial contribution from the mode at $\sim 3100\text{ cm}^{-1}$, which well matches the mode of the N–H stretching motion. Moreover, this signal is enhanced significantly when directly exciting the N–H stretching mode (**Fig. 5c** inset). Furthermore, the N–H bond length increases with the growth of excitation energy (Supplementary Fig. S20), which can be attributed to the thermal expansion of organic cation due to IVER.

In our previous calculations, we only considered the effect of static N–H stretching on the bandgap. This may not be adequate to describe the influence of N-H stretching motion on the bandgap of halide perovskites, as pointed out in Comment 1) raised by Reviewer #1. Thus, we further employed ab initio molecular dynamics (AIMD) to verify the non-trivial effect of organic and inorganic sublattice coupling. Details of the response can be referred to Comment 1) raised by Reviewer #1.

Comment 3) The mechanism of down-conversion of the organic cation vibrations into long-range phonons in the inorganic lattice is also unclear. While the organic vibrations are localized (and not phonons) and have no well-defined momentum, the phonons in the lattice have. How is momentum conservation not violated in such a process?

Response: We thank the reviewer for raising this important point. We would like to highlight that the vibrational motion of organic cation does not decay directly into the phonons of inorganic octahedra. In fact, this “phonon down-conversion” process is different from the conventional anharmonic phonon-phonon decay process widely reported in most semiconductors, in which a high-energy phonon decays into two or more low-energy daughter phonons while preserving both energy and momentum. This phonon down-conversion involves two successive processes: the intramolecular vibrational energy redistribution within the PEA cation, followed by the heat transfer from the thermalized organic PEA cation to the inorganic PbI_6 octahedra through vibrational coupling (*Nat. Commun.*, **10**, 482 (2019), *Nanoscale*, **12**, 9661 (2020)). For the former, it involves the cascade energy down-conversion process through all the intermediate phonon modes starting from the excited highest-lying N-H stretching mode to the lowest-lying modes of PEA cation by anharmonic vibrational coupling. The duration of this process is in general longer than the lifetime of any activated intermediate modes. For the latter, it is governed by the vibrational states overlap and coupling between these two sublattices (*Nat. Commun.*, **10**, 482 (2019), *Nat. Mater.*, **12**, 410 (2013)). Typically, this process is much longer and takes place on the order of tens of picosecond compared to those of intramolecular vibrational energy relaxation and conventional anharmonic phonon decay that normally occur within several picoseconds (*Phys. Rev. B*, **57**, 12847 (1998), *Phys. Chem. Chem. Phys.*, **18**, 27051 (2016)).

In view of this feedback, we have cautioned our statement of this phonon down-conversion process and made necessary changes to the main text, as shown below:

At line 32 on page 2 of the main text:

Further evidence of the mechanical coupling **between these two sublattices** is revealed through the slow **heat transfer** process ~~between their vibrational motions~~, where the resultant lattice tensile strain launches the coherent longitudinal acoustic phonons.

At line 72 on page 4 of the main text:

The former contributes to the modified lattice degree of freedom of the inorganic sublattice which is associated with a decrease of the optical bandgap arising from the reduced out-of-plane distortion of inorganic sublattice, whereas the latter contributes to the slow **heat transfer** process, resulting in the generation of thermally driven-lattice strain that launches the soft coherent longitudinal acoustic phonons (CLAPs).

At line 153 on page 8 of the main text:

Regime ③ can be assigned to **the slow heat transfer** process where the vibrational energy of the thermalized PEA cation is transferred to the phonon modes of PbI_6 octahedra that will eventually reach a quasi-equilibrium state amongst themselves. **In this process, the** population of high-energy phonon modes is reduced which is associated with the population increase of low-energy phonon modes. Consequently, the whole lattice will be heated up, resulting in a lattice expansion which is associated with the increase of exciton transition energy, thereby generating a band-filling-like TA spectrum.³² The thermal lattice expansion-induced exciton energy increase is consistent with the temperature dependence of the optical bandgap of $(\text{PEA})_2\text{PbI}_4$, which possesses a positive bandgap deformation potential.^{7,33} Similar **slow heat transfer** process from relaxation of vibrational motion of the organic lattice to that of the inorganic lattice has been reported in MAPbI_3 films³² and colloidal CdSe nanocrystals³⁴. In contrast with

conventional phonon relaxation process (such as intramolecular vibrational energy relaxation³⁵ and phonon-phonon equilibration in most semiconductors^{19,36}) due to anharmonic phonon-phonon interactions which in general take place within several picoseconds³⁷, this slow heat transfer process arising from interactions between different sublattices is two orders of magnitude slower. This may be because (1) these two sublattices are connected via a much weaker hydrogen bonding and electrostatic interactions instead of the stronger covalent or ionic bonding, and (2) the overlapping vibrational density of states and the coupling between these two sublattices are relatively weak given that there is a large energy gap between the phonon modes of these two sublattices due to their large mass difference (Supplementary Fig. S5). Consequently, a much longer time will be needed for these two sublattices to reach a quasi-equilibrium state. This slow heat transfer can be further verified from the extended duration of regime ③ with increasing pump fluence (Supplementary Fig. S5).

At line 331 on page 17 of the main text:

Next, we investigate the CLAPs-induced oscillation kinetics after phonon down-conversion process (*i.e.*, intramolecular vibrational energy redistribution within PEA cation followed by heat transfer from thermalized PEA cation to PbI₆ octahedra) in regime ④. After subtracting the non-oscillatory components which are fitted with multiple exponential decay functions, we obtain the time-domain of CLAPs-induced beating map, as shown in **Error! Reference source not found.a**.

At line 342 on page 18 of the main text:

Furthermore, with increasing pump fluence, the CLAPs-induced oscillation is delayed which can be attributed to an even more inefficient heat transfer such that the lattice displacement takes more time to reach its maximum amplitude (**Fig. 4b**).

At line 386 on page 20 of the main text:

For the former, the nonlinear interaction potential results in off-diagonal elements in the vibrational Hamiltonian matrix which mixes the high-energy phonon modes of PEA cation with the low-energy phonon modes of PbI₆ octahedra, creating new pathways for energy relaxation and dissipation. This mechanical coupling agrees well with the observed slow heat transfer process in regime ③.

At line 422 on page 21 of the main text:

Compared to off-resonant pump, resonant pump of the PEA cation results in modification of both electronic and lattice degrees of freedom of PbI₆ octahedra by changing its structural distortion. The weak coupling between these two sublattices leads to a slow heat transfer process, in which the resultant lattice strain launches the coherent longitudinal acoustic phonons.

And in SI, we have added the following notes to Supplementary Fig. S5:

Fig. S5 **a** Raman spectrum of $(\text{PEA})_2\text{PbI}_4$ at 77 K. Inset shows the zoom-in of the Raman spectrum that is dominated by the vibrational motion of PEA cation. **b** Fluence-dependent TA kinetics of $(\text{PEA})_2\text{PbI}_4$ films monitored at P1 with pump at $3.3\mu\text{m}$. **c** Duration of regime ③ as a function of pump fluence (scatters) and the corresponding curve-fit (red curve) using $y = y_0 + A \exp\left(-\frac{I}{I_0}\right)$, where A , y_0 and I_0 are constants.

As reported in previous works (*Nat. Mater.*, 18, 349 (2019), *Phys. Rev. Mater.*, 2, 034001 (2018)), the low-frequency phonon modes ($< \sim 150 \text{ cm}^{-1}$) are dominated by the vibrational motion of PbI_6 octahedra with the mode located at $\sim 135 \text{ cm}^{-1}$ exhibiting mixed contributions from PEA cations and PbI_6 octahedra whereas the high-frequency phonon modes ($> 200 \text{ cm}^{-1}$) are governed by the vibrational motion of PEA cations. The vibrational coupling between these two sublattices is thus relatively weak.

Note that in **Fig. S5a**, the phonon down-conversion process involves two successive processes: intramolecular energy redistribution within the PEA cation followed by the heat transfer from organic PEA cation to inorganic PbI_6 octahedra through anharmonic coupling (*Nat. Commun.*, 10, 482 (2019), *Nanoscale*, 12, 9661 (2020)). For the former, it involves sequential energy down-conversion through all the intermediate phonon modes starting from the excited highest-lying mode (*i.e.*, N-H stretching motion) to the lowest-lying mode. The duration of this process is in general longer than the lifetime of any activated intermediate modes. For the latter, it involves the heat transfer from the phonon modes of organic cation to those of inorganic octahedra which is governed by the overlapping vibrational density of states and the coupling between the organic and inorganic sublattices (*Nat. Commun.*, 10, 482 (2019), *Nat. Mater.*, 12, 410 (2013)). Nevertheless, the large mass difference which results in a large energy difference of the phonon modes as well as their relatively weak coupling though hydrogen bonding and electrostatic interaction between these two sublattices lead to a significant weak mechanical coupling, thereby slow heat transfer.

This inefficient mechanical coupling results in slow heat transfer which is evident from the increase of duration time of regime ③ with pump fluence.

Comment 4) Moreover, the authors observe the generation of coherent strain waves. What is the suggested mechanism through which the relatively independent motion of organic moieties leads to such a long-range coherence? Even if one assumes that the MIR pump imposes coherence between the motion of the organic cation, can such a coherence last for several tens of picoseconds to enable the generation of coherent acoustic phonons?

Response: We thank the reviewer for raising this question. As mentioned in the previous works by Thomsen *et al.* (*Phys. Rev. B*, **34**, 4129-4138 (1986), *Ultrasonics*, **56**, 21 (2015)) and our recent work (*Sci. Adv.*, **8**, eabq1971 (2022), *Chem. Rev.*, **123** 8154 (2023)), the CLAPs or the propagating strain pulse occurs when the lattice is dynamically deformed under the stress. We can understand the generation of CLAPs based on a phenomenological one-dimensional elastic wave motion along z axis, which gives rise to (*Appl. Phys. Lett.*, **66**, 1190 (1995), *Phys. Rev. B*, **34**, 4129 (1986)):

$$\frac{\partial}{\partial \rho} \rho \frac{\partial u}{\partial t} - \frac{\partial}{\partial z} C \frac{\partial u}{\partial z} = \frac{\partial \sigma}{\partial z} \quad (3)$$

where ρ , u , σ are the mass density, lattice displacement, and source of stress, respectively. And $C = \rho v^2 \eta$ is the elastic constant, v is the sound velocity and $\eta = \frac{\partial u}{\partial z}$ is the strain. Depending on the source of the stress σ , the generation mechanism of CLAPs thus varies. Basically, there are mainly three sources of the stress: (i) thermoelastic (TE) effect, (ii) deformation potential (DP) interaction, and (iii) inverse piezoelectric interaction. The first (i) occurs when the photoexcited carriers transfer their excess energies to the lattice via electron-phonon interaction as they relax down to the band edge, generating a fast increase in the lattice temperature, which yields the resultant TE stress (*Phys. Rev. B*, **34**, 4129-4138 (1986), *Ultrasonics*, **56**, 21-35 (2015)). The second (ii) takes effect when the lattice equilibrium is broken and deformed by the change in electronic distribution after photoexcitation, which in turn modifies the band structure, inducing electronic stress. The last (iii) is present when the intrinsic built-in electric field of a non-centrosymmetric material is screened by the photoexcited charge carriers, which changes the equilibrium state of the lattice through the inverse piezoelectric effect, leading to the inverse piezoelectric stress. These three effects have been discussed in our previous work where we explicated the presence of first two contributions in photoexcited (PEA)₂PbI₄ single crystals (*Sci. Adv.*, **8**, eabq1971 (2022)). However, these two mechanisms are a bit different from our case here, which does not involve direct laser-matter interactions. We can first exclude the DP interaction since there is no electronic excitation under MIR pump. However, for the TE effect, to be more precise, it does not apply to our case as well since there is no electronic excitation. Nevertheless, the consequential effect of MIR pump is the fast heating of the whole lattice because of sequential phonon population process, *i.e.*, heat transfer from the organic cation to inorganic octahedra, which in turn generates the TE stress. At this point, it is similar to the conventional TE effect for the above-bandgap pump. Thus, we agree with Comment 3) by Reviewer #3 that the generation mechanism of the observed CLAPs is TE effect.

In view of this concern, we have cautioned our statement *at line 367 on page 19 of the main text*, as shown below:

It is important to highlight that the mechanism of CLAPs' generation in this work is different from **most of the reports in which CLAPs take place after direct light-matter interactions** (*Phys. Rev. B*, **34**, 4129 (1986), *Ultrasonics*, **56**, 21 (2015)). **From these reports, three mechanisms are identified: the thermoelastic effect due to fast increase of the lattice temperature^{7,56,57}, the deformation potential interaction resulting from modification of the electronic transition due to carrier generation^{7,57}, and the inverse piezoelectric effect owing to a change of the lattice's equilibrium state when the surface built-in electric field is varied after carrier generation^{47,57}.** Nevertheless, for our

case which does not involve any electronic transitions, the CLAPs are launched due to the heat transfer from the organic cation to the inorganic octahedra, which in turn generates the thermoelastic stress.

Meanwhile, we have also modified the statement *at line 181 on page 9 of the main text*, as shown below:

As demonstrated below, the phonon coherence in regime ② with resonant MIR pump is distinct from that with off-resonant IR pump and that in regime ④ arises from the thermoelastic effect due to the increase of lattice temperature by heat transfer from the thermalized organic cations to the inorganic octahedra, thereby providing further evidence of the organic and inorganic sublattice coupling.

Reviewer #3:

General comments: The authors report on a study of the 2d hybrid perovskite (PEA)₂PbI₄ under mid-infrared resonant excitation of the organic cations, probing energy transfer between the organic and inorganic lattice. As they note a prior Nature Comm paper by Guo (reference 30) has carried out a similar experiment in the 3d halide perovskites, observing only slow coupling via thermal effects. Here a more complex dynamical response is observed spanning a wide range of time-scales. I think the results are interesting and will be of interest to the broader community interested in the perovskites although I have a number of questions and comments.

Response: We thank the reviewer for his/her valuable time to critically review our manuscript and greatly appreciate his/her feedback to help us strengthen our manuscript. We have carefully considered the referee's comments and below are our replies:

Comment 1) How is it known that the 3.3 μm pump only couples initially to the organic lattice? Are there higher order phonon resonances from the inorganic lattice that could lie there? How about an impulsive stimulated Raman scattering response arising from the sub-gap excitation of the inorganic lattice? This would give rise to the linear in fluence response and could represent a more direct mechanism for generating the observed phonon responses.

Response: We thank the reviewer for this constructive feedback. The 3.3 μm pump couples to the vibrational motion of both the inorganic PbI₆ octahedra and organic PEA cations. For the former, it leads to generation of coherent optical phonons (COPs), whereas for the latter it results in depletion of MIR absorption which in turn reduces the exciton energy that leads to distinct TA features.

Firstly, there is negligible electronic coupling between inorganic sublattice and 3.3 μm pump because the carrier generation rate with multiple-photon absorption is much lower. As has been widely demonstrated, the band-edge electronic density of states of lead iodine perovskites including (PEA)₂PbI₄ are dominated by the orbitals from Pb and I atoms (*Appl. Phys. Lett.*, **104**, 063903 (2014), *Phys. Rev. B*, **89**, 155204 (2014), *Chem. Rev.*, **123** 8154 (2023)). The optical bandgap of (PEA)₂PbI₄ at 77 K (~2.4 eV) is more than 6 times the MIR pump energy of 3.3 μm (~0.38 eV), suggesting that seven-photons must be absorbed simultaneously for electronic excitation. However, the carrier generation rate in this case is negligible, as shown below.

In general, we can estimate the carrier generation rate when absorbing n -photons simultaneously as (*Chem. Rev.*, **123**, 8154 (2023)):

$$G_{np} = \frac{\alpha_n I^n}{n \hbar \omega} \quad (4)$$

where α_n , I , $\hbar\omega$ are respectively the n -photon absorption coefficient, pump power, and pump energy, respectively. For our case with pump fluence of ~2 mJ cm⁻² and the reported champion $\alpha_5 \sim 10^{22}$ cm⁷ GW⁻⁴ (*Adv. Mater.*, **32**, 2002972 (2020)), the carrier generation rate for five-photon absorption is estimated as ~10⁶ cm⁻³ s⁻¹, which is 9 orders lower than that in a typical TA measurement. On the other hand, as is well known, the higher the order of multiple-photon absorption, the lower the nonlinear optical absorption coefficient. This suggests the carrier density for seven-photon absorption will be even lower. This is further verified from our TA measurement exhibiting distinct TA features compared to those by multiphoton absorption due to the band-filling effect, as displayed in Supplementary Fig. S9 below. Thus, there is negligible electronic excitations with 3.3 μm pump.

Fig. S9 Comparison of 2D contour-plot of TA spectrum, representative TA spectrum at different delay times and TA kinetics monitored at P1 of $(\text{PEA})_2\text{PbI}_4$ films obtained with resonant pump using an MIR laser pulse at $3.3 \mu\text{m}$ (**a**, **b**, and **c**) and with off-resonant pump using an IR laser pulse at $1.5 \mu\text{m}$ (**d**, **e**, and **f**). Note that **a**, **b** and **c** are replots of Figure 1 of the main text.

Secondly, it is unlikely that high-order phonon resonance of inorganic octahedra close to $3.3 \mu\text{m}$ plays an important role. This is because the dominant vibrational motion of inorganic PbX_6 octahedra (*Phys. Chem. Chem. Phys.*, **18**, 27051 (2016), *J. Phys. Chem. Lett.*, **7**, 1 (2015), *J. Phys. Chem. Lett.*, **5**, 279 (2013), *Nat. Mater.*, **18**, 349 (2019), *Phys. Rev. Mater.*, **2**, 034001 (2018)) occurs at the low-frequency phonon modes ($< \sim 150 \text{ cm}^{-1}$). This suggests that a phonon overtone of more than 20-orders is needed for a resonance phonon coupling under pump of $3.3 \mu\text{m}$ ($\sim 3030 \text{ cm}^{-1}$). Nevertheless, the two-phonon overtone with frequency between $\sim 40 \text{ cm}^{-1}$ and $\sim 100 \text{ cm}^{-1}$ is absent in our TA measurement, which has a detection resolution up to $\sim 200 \text{ cm}^{-1}$.

In fact, the $3.3 \mu\text{m}$ pump couples to the inorganic sublattice via launching the COPs through the ISRS as well. However, compared to the off-resonant below-bandgap excitation, the driving force for the $3.3 \mu\text{m}$ pump has a non-trivial contribution from real excitation due to the absorption from PEA cation. As discussed in our manuscript, within the two-band process dominated Raman tensor and the fast decay rate of the coupled carriers, the driving force consists of contributions from both virtual and real excitations (*Phys. Rev. B*, **65**, 144304 (2002), *Adv. Mater.*, **33**, 2006233 (2021)):

$$F(\Omega) = -C\Xi \left[\frac{\partial \epsilon_R}{\partial \omega} + 2i \frac{\epsilon_{\text{Im}}}{\Omega} \right] I(\Omega) \quad (5)$$

where C is a prefactor, ϵ_R and ϵ_{Im} are, respectively, the real and imaginary parts of the dielectric constant, Ξ is the deformation potential constant. For the below-bandgap pump with a vanishing ϵ_{Im} which is the case for our off-resonant pump at 1.2 , 1.5 and $2.1 \mu\text{m}$, the driving force is governed by the first term in the bracket of Eq. (5). In contrast, for the resonant pump at $3.3 \mu\text{m}$, there is a significant contribution from the real excitation due to the MIR absorption governed by N–H stretching motion. And this significant real excitation leads to an enhanced linear response of vibrational amplitude with respect to the pump fluence, as shown in **Fig. R6**. On the other hand, the

presence of non-trivial real excitation with resonant pump would also explain well the phase differences of the activated phonon modes under resonant pump compared to those under off-resonant pump. In light of the pump fluence dependence, we would like to point out that as long as the phonon perturbation is in the linear range, the phonon-induced absorbance modulation amplitude will linearly depend on the pump fluence since the driving force is proportional to the pump power.

Fig. R6 FFT amplitude of M2 mode as a function of pump fluence for off-resonant IR pump at 1.5 μm (black squares) and resonant MIR pump at 3.3 μm (red circles). The red lines are the linear fits.

In view of this feedback, we have added the following statements *at line 319 on page 16 of the main text*, as shown below:

Other than $(\text{PEA})_2\text{PbI}_4$, we also observe a decrease of the optical bandgap associated with phonon frequency shifts of PbI_6 octahedra after resonant excitation of organic cation in 2D hexylammonium lead iodide $(\text{HA})_2\text{PbI}_4$ perovskite (Supplementary Fig. S10), further demonstrating the organic and inorganic sublattice coupling in hybrid halide perovskites. Nevertheless, compared to $(\text{PEA})_2\text{PbI}_4$, the phonon frequency shifts in $(\text{HA})_2\text{PbI}_4$ are smaller, indicating a weaker organic and inorganic sublattice coupling.

Note that the COPs' generation mechanisms for both off-resonant and resonant pumps are ISRS. Nevertheless, the driving force of the lattice displacement for the former is governed by the virtual excitation (*i.e.*, the first term in the bracket of Eq. (5)) whereas for the latter, there is a significant contribution from the real excitation (*i.e.*, the second term in the bracket of Eq. (5)) due to the MIR absorption governed by N–H stretching motion. This nontrivial contribution from the real excitation leads to an enhanced linear response of vibrational amplitude to the pump fluence (Supplementary Fig. S11). The presence of this non-trivial real excitation with resonant pump would also well explain the phase differences of the activated phonon modes under resonant pump compared to those under off-resonant pump. Moreover, if the phonon perturbation is in the linear range, the phonon-induced absorbance modulation amplitude will linearly depend on the pump fluence since the driving force is proportional to the pump power. On the other hand, resonant coupling between the high-order overtone and the 3.3 μm MIR pump pulse ($\sim 3030\text{ cm}^{-1}$) is unlikely given that vibrational frequencies of the inorganic sublattice are below $\sim 150\text{ cm}^{-1}$ such that a phonon overtone of more than 20-orders would be required. (*Nat. Mater.*, **18**, 349 (2019), *Phys. Rev. Mater.*, **2**, 034001 (2018)). Similarly, electronic excitation from multiple-photon absorption (*i.e.*, needing simultaneous absorption of seven-photons) would also be negligible. Thus, the MIR pump pulse couples to the vibrational motion of both inorganic PbI_6 octahedra and organic PEA cation. For the former, it leads to generation of COPs, whereas

for the latter, it results in the depletion of MIR absorption which in turn reduces the exciton energy that leads to distinct TA features.

Meanwhile, we have added Supplementary Fig. S11 to the SI, as shown below:

Fig. S11 FFT amplitude of M2 mode as a function of pump fluence for off-resonant IR pump at 1.5 μm (black squares) and resonant MIR pump at 3.3 μm (red circles). The red lines are linear fits.

Comment 2) The very fast response seen in regime 1 is attributed to an optical stark effect e.g. driven directly by the light field. If such a coupling exists and modulates the bands associated with the inorganic lattice, does this not then directly couple to the inorganic lattice through e.g. deformation potential coupling? This would provide another mechanism that should be discussed in the answer to question 1 above.

Response: We thank the reviewer for raising this concern. The inorganic sublattice does couple to the laser pulse through deformation potential, which results in the generation of coherent optical phonons observed. The discussion on the generation mechanism of these activated optical phonons can be referred to Comment 1) above.

Comment 3) The slower response in regime 3 as I understand it is arising from a pure heating of the inorganic lattice which leads to lattice expansion mediated by acoustic phonons. Presumably the time-scale observed here is just the film thickness divided by sound velocity? It is stated that "It is important to highlight that the mechanism of CLAP generation in this work is different from previously reported mechanisms of thermoelastic effect, deformation potential interaction, or inverse piezoelectric effect". Is not this response exactly a thermoelastic effect? As soon as coupling occurs to the inorganic lattice it heats and this follows an acoustic response as originally investigated by e.g. Thomsen et al. decades ago (Phys. Rev. B, 34, 4129 (1986)). Are the authors claiming the mechanism for the CLAP response is distinct from this?

Response: We thank the reviewer for raising this important point. The reply to this comment is similar to Comment 4) raised by Reviewer #2. Kindly refer to the response above on page 15.

Comment 4) Overall the big picture on why these results are important is lacking in the current version. The paper concludes with some comments about how this would impact applications with respect to next generation nonlinear phononics which is unclear. More importantly, it is unclear how the results of this study elucidate our understanding of the optoelectronic functionality of these materials. In the original Guo paper the story was somehow that all of the unique properties of MAPbI₃ arise from the inorganic lattice, given the weak coupling. Here there is a strong coupling - so what does this tell us about the key unknown aspects of the 2D perovskites? Without a good answer to this this paper is of interest to specialists in the field but does not truly impact our understanding of these materials.

Response: We thank the reviewer for raising this constructive feedback to strengthen our manuscript. The interactions between the organic and inorganic lattices can have profound effects on the optoelectronic properties of 2D halide perovskites. In our manuscript, we demonstrated the presence of both *electronic* and *mechanical couplings* between the organic and inorganic sublattices of 2D halide perovskites.

Strong electronic coupling suggests that the vibrational motion of the organic cation has a large impact on the electronic and vibrational properties. It was reported in previous works that the rotational motion of the organic cation can screen the excitonic feature, thereby reduce the exciton binding energy of halide perovskites (*J. Phys. Chem. C*, **118**, 11566 (2014)); as well as the reduction of the biexciton binding energy due to the vibrational motion of organic cations-induced dynamic disorder (*Phys. Rev. Mater.*, **2**, 034001 (2018)). Meanwhile, by varying the vibrational motion of organic cation (e.g., via temperature), the electronic transition as well as the exciton-phonon coupling (and therefore polaron formation) can thus be modified. This will allow tuning of the optoelectronic properties of 2D halide perovskites as demonstrated in a recent work by Gallop *et al.* (10.1038/s41563-023-01723-w). This opens exciting new possibilities for ultrafast control of the optoelectronic properties of 2D halide perovskites via modulating the local motion of the organic sublattice.

On the other hand, the *strong vibrational coupling* between the low-energy and high-energy modes can produce high-energy vibrational combination states composed of mixed localized and delocalized nuclear motions, having relatively low vibrational quantum numbers. This would then provide additional efficient relaxation pathways for hot carriers. As for weak *mechanical coupling* which means a small sound velocity, it indicates the cross-plane thermal conductivities of 2D layered halide perovskites are intrinsically low. This agrees well with previous reported ultralow thermal conductivities (~0.099–0.125 W/m K) that are insensitive to the length and type of the organic cation in 2D halide perovskites (*Nano Lett.*, **20**, 3331 (2020), *ACS Appl. Mater. Interfaces*, **12**, 53705 (2020)). Thus, our work is essential to not only understanding the fundamental photophysics but also relevant for engineering ultrafast control of 2D halide perovskites-based devices. We believe that our work will attract broad interests of scientists from diverse fields spanning ultrafast optical spectroscopy to materials engineering as well as optoelectronic applications, which meets high standards of *Nature Communications*.

In view of this feedback, we have modified the statement at line 385 on page 20 of the main text, as shown below:

This mechanical coupling agrees well with the observed **slow heat transfer** process in regime ③. **Such weak mechanical coupling which indicates a small sound velocity, indicates that the cross-plane thermal conductivities of 2D layered halide perovskites are intrinsically low and insensitive to the type of the organic moiety, which agrees well with previous results (*Nano Lett.*, **20**, 3331 (2020), *ACS Appl. Mater. Interfaces*, **12**, 53705 (2020)).** As for the latter, the interaction between the electronic degree of freedom of the PEA cation and the vibrational motion of the PbI₆ octahedra will modify the potential energy surface, and thereby the density of states, vibrational anharmonicity and dispersion of PbI₆ octahedra.⁶⁰ This electronic coupling is again consistent with the observed modification of the activated phonon modes (*i.e.*, frequency and dephasing time) of PbI₆ octahedra in regime ②. On the other hand,

this electronic coupling is in line with the observed reduction of optical bandgap, which is likely due to the varied distortion of PbI_6 octahedra, as demonstrated below. Similar strong coupling was also reported in which vibrational motion of organic cation leads to screening of excitonic features (*J. Phys. Chem. C*, **118**, 11566 (2014)) and reduction of biexciton binding energy (*Phys. Rev. Mater.*, **2**, 034001 (2018)). Such strong coupling could offer us an exciting lever to exert ultrafast control over the optoelectronic properties of 2D halide perovskites by simply modulating the local motion of the organic sublattice (*Nat. Mater.*, **23**, 88–94 (2024)).

As for the implications in the nonlinear phononics, the coupling between the organic and inorganic sublattices in 2D halide perovskites revealed by our work would provide a unique route to exert ultrafast control over their electronic transitions dominated by inorganic PbI_6 octahedra by manipulating the vibrational motions of organic PEA cation, as demonstrated in a recent work by Gallop *et al.* (doi: 10.1038/s41563-023-01723-w). And this is different from previously mentioned nonlinear phononic control in which both the coupled Raman and Infrared modes arise from vibrational motions of octahedra sublattice (*Nat. Phys.*, **7**, 854 (2011)). Thus, we claim that our findings may find important applications in next-generation nonlinear phononics for controlling properties of the materials via coherent phonons.

In view of the reviewer's feedback, we have cautioned our statement *at line 424 on page 22 of the main text*, as shown below:

Our work injects fresh insights into the organic and inorganic sublattice coupling that would be important for clarifying additional energy relaxation pathway for the photoexcited energetic carriers as well as **enabling ultrafast control over the optoelectronic properties of 2D halide perovskites through nonlinear phononics** (*Nat. Mater.*, **23**, 88–94 (2024)).

REVIEWERS' COMMENTS

Reviewer #1 (Remarks to the Author):

The revised manuscript is significantly improved, as the authors completely changed the theoretical approach to model the systems. AIMD are indeed better suited to support the observed experimental band-gap change. Yet, I am still not convinced that such a small variation (i.e., of 0.1 meV) is within the accuracy of the method. A phrase could be added to caution the reader about this.

Regarding my comment regarding the van der Waals forces: I appreciate the detailed explanation, and I now understand what the authors mean. However, the phrase in the introduction in the manuscript remains confusing. What is actually written implies that the organic and inorganic lattice is hold together by vdw forces. The authors should rephrase to make clear that the organic-organic interaction between adjacent layers are mainly vdw.

Reviewer #2 (Remarks to the Author):

The authors have addressed all the comments I had raised previously and the manuscript can be considered for publication.

I however lack the expertise to judge the level and suitability of theoretical model presented in the manuscript, in the light of comments from Reviewer 1.

Reviewer #3 (Remarks to the Author):

I apologize for my late reply on this - I think the authors have done a very careful job of replying to all comments I recommend publication.

- Reviewers' comments are in brown
- Our responses are in black
- The sentences revised or added in the manuscript are highlighted in blue

Reviewer #1:

Comment 1) The revised manuscript is significantly improved, as the authors completely changed the theoretical approach to model the systems. AIMD are indeed better suited to support the observed experimental band-gap change. Yet, I am still not convinced that such a small variation (i.e., of 0.1 meV) is within the accuracy of the method. A phrase could be added to caution the reader about this.

Response: We greatly appreciate the reviewer's valuable time in reviewing our manuscript as well as his/her feedback. Indeed, there is a difference between the experimental results and theoretical calculations. In fact, we have considered two possibilities: one is from the smearing effects due to scattering from the grain boundaries and defects, another is because only a ~3% of PEA cation is excited in our experiments in contrast with all the PEA cations being excited in our calculations. Nonetheless, these two effects are discussed in the main text and Supplementary Note 3 (shown below) in our previous manuscript.

Supplementary Note S3

We note that our simulated result (~10 meV) is nearly 2 orders of magnitude larger compared to the experimental results (~0.1 meV). This is likely because we excite all the PEA cation in our simulations while only a small fraction of PEA cation is excited in the sample. To verify this, we first estimate the number of excited PEA in the sample using:

$$n = \frac{P\alpha}{hv} \quad (1)$$

where P is the pump power, α is the IR absorption coefficient, and hv is the pump energy. For typical pump power of ~1 mJ cm⁻³, α of ~2×10³ cm⁻³, n is estimated as 3.3×10²⁰ cm⁻³.

And the number of PEA cation N in 1 cm³ is calculated using:

$$N = 8 \frac{\rho}{M} N_A \quad (2)$$

where ρ is the mass density (~2.55 g cm⁻³), M is the molar mass density (959.19 kg mol⁻¹), N_A is the Avogadro's number. Here, N is estimated as ~1.3×10²². The fraction of excited PEA cation is then estimated as n/N ~2.6×10⁻². Assuming a linear relationship between the optical bandgap and pump power which holds for our case based on our experimental results (Figure 2c of the main text), the estimated reduced optical bandgap is thus reasonable.

In view of the reviewer's feedback, we have added on *page 23* of the main text a caveat on the potential limitation on the accuracy of our AIMD simulations, as shown below:

Note that the reduced bandgap (~0.1 meV) in our measurement is nearly 2 orders of magnitude smaller compared to our simulated result (~10 meV). This may be due to the smearing effects from the grain boundaries and defects or because we excite all the PEA cations in our simulations whereas only a small fraction of PEA cations is excited in the sample (Supplementary Note S3). **Further studies may be needed to clarify the differences between the experimental results and theoretical calculations.**

Comment 2) Regarding my comment regarding the van der Waals forces: I appreciate the detailed explanation, and I now understand what the authors mean. However, the phrase in the introduction in the manuscript remains confusing. What is actually written implies that the organic and inorganic lattice is hold together by vdw forces. The authors should rephrase to make clear that the organic-organic interaction between adjacent layers are mainly vdw.

Response: We thank the reviewer for the feedback. We have clarified our description on this vdW force on *page 3* of the main text, as shown below:

These octahedra are sandwiched by bilayers of bulkier organic spacers (Fig. 1a) that are *in turn* held together by van der aals forces^{5,6} similar to 2D van der Waals heterostructures⁷ and MXenes⁸.

Reviewer #2:

Comment: The authors have addressed all the comments I had raised previously, and the manuscript can be considered for publication. I however lack the expertise to judge the level and suitability of theoretical model presented in the manuscript, in the light of comments from Reviewer 1.

Response: We greatly appreciate the reviewer for his/her valuable time for reviewing our manuscript. As acknowledged by Reviewer #1, AIMD simulations is a more reliable and powerful approach to simulate the bandgap kinetics after exciting PEA cation. Though there are some deviations between experimental results and theoretical calculations which may be because only a fraction of PEA cations is excited in our experiments, our approach does demonstrate a clear correlation between the reduced optical bandgap after exciting PEA cation. Moreover, this method has also been employed to demonstrate the organic cation vibration-induced promotion of electronic resonance (*i.e.*, reduced bandgap) in lead bromide perovskites in a recent work (*Nat. Mater.*, **23**, 88–94 (2024)). Thus, to our knowledge, we think that our AIMD simulations is the current most reliable approach to verify our experimental results. In light of the feedback, we have added a caveat on the potential limitation on the accuracy of our AIMD simulations, as shown in Comment 1) raised by Reviewer #1.

Reviewer #3:

Comment: I apologize for my late reply on this - I think the authors have done a very careful job of replying to all comments I recommend publication.

Response: We are delighted with the reviewer's comments and all the valuable feedback. We would like to thank him/her for his/her time to help strengthen our manuscript for publication.